# Rapid chromosome evolution and acquisition of thermosensitive stochastic sex determination in nematode androdioecious hermaphrodites

Kohta Yoshida [1,2,3] ✉, Hanh Witte[1], Ryo Hatashima [4], Simo Sun [5], Taisei Kikuchi [5], Waltraud Röseler[1] & Ralf J. Sommer [1] ✉

The factors contributing to evolution of androdioecy, the coexistence of hermaphrodites and males such as in *Caenorhabditis elegans*, remains poorly known. However, nematodes exhibit androdioecy in at last 13 genera with the predatory genus *Pristionchus* having seven independent transitions towards androdioecy. Nonetheless, associated genomic architecture and sex determination mechanisms are largely known from *Caenorhabditis*. Here, studying 47 *Pristionchus* species, we observed repeated chromosome evolution which abolished the ancestral XX/XO sex chromosome system. Two phylogenetically unrelated androdioecious *Pristionchus* species have no genomic differences between sexes and mating hermaphrodites with males resulted in hermaphroditic offspring only. We demonstrate that stochastic sex determination is influenced by temperature in *P. mayeri* and *P. entomophagus*, and CRISPR engineering indicated a conserved role of the transcription factor TRA-1 in *P. mayeri*. Chromosome-level genome assemblies and subsequent genomic analysis of related *Pristionchus* species revealed stochastic sex determination to be derived from XY sex chromosome systems through sex chromosome-autosome fusions. Thus, rapid karyotype evolution, sex chromosome evolution and evolvable sex determination mechanisms are general features of this genus, and represent a dynamic background against which androdioecy has evolved recurrently. Future studies might indicate that stochastic sex determination is more common than currently appreciated.

In animals, several distinct sexual systems exist, in which male and female functions are distributed differently across the individuals of a population (Fig. 1a). Specifically, species with simultaneous or sequential hermaphroditism are found in many animal lineages, similar to obligate outcrossing species (dioecy). In contrast, androdioecy, the coexistence of hermaphrodites and males is sparsely found across animals and considered to be an evolutionarily intermediate state[1–3] (Fig. 1a). Notwithstanding, the nematode *Caenorhabditis elegans* is an androdioecious hermaphrodite and its success as model organism is in large parts due to the occurrence of males that allow outcrossing[4].

[1]Department for Integrative Evolutionary Biology, Max Planck Institute for Biology Tübingen, Tübingen, Germany. [2]Ecological Genetics Laboratory, National Institute of Genetics, Mishima, Japan. [3]Department of System Pathology for Neurological Disorders, Brain Research Institute, Niigata University, Niigata, Japan. [4]School of Life Science and Technology, Institute of Science Tokyo, Meguro-ku, Tokyo, Japan. [5]Department of Integrated Biosciences, Graduate School of Frontier Sciences, The University of Tokyo, Chiba, Japan. ✉e-mail: kyoshida@bri.niigata-u.ac.jp; ralf.sommer@tuebingen.mpg.de

While classic theoretical studies provide strong support for the evolution of gynodioecy (the coexistence of hermaphrodites and females), which represents yet another hermaphroditic system (Fig. 1a), the theory of the evolution of androdioecy is limited[2,5–8]. Similarly, empirical studies for evolutionary transitions towards androdioecy are relatively sparse[6]. Nonetheless, in contrast to what was thought until 10–15 years ago, recent studies in nematodes revealed that androdioecy has independently evolved in at least 13 genera[2]. In particular, the predatory genus *Pristionchus* contains eight androdioecious species that result from at least seven independent transitions[9,10]. Thus, androdioecy has evolved many times independently in nematodes. However, the genomic and molecular mechanisms underlying androdioecy and associated sex determination systems are largely known from *Caenorhabditis* with additional recent work in some *Bursaphelenchus* species[11]. This study aims to enhance the understanding of the mechanisms that are associated with the recurrent evolution of androdioecy by focusing on the genus *Pristionchus* with its many independent transitions towards androdioecy.

In general, self-fertilizing nematode hermaphrodites can outcross to males but not to other hermaphrodites. Therefore, males are crucial for facultative outcrossing, which is similar to other androdioecious organisms[12]. All known hermaphrodites are derived from gonochoristic (dioecious) ancestors with separate male and female individuals. In *Caenorhabditis*, gonochoristic and androdioecious species have six chromosomes and a genetic sex determination (GSD) system that relies on the X:A (X chromosome–autosome) ratio. Specifically, in *C. elegans*, XX animals become hermaphrodites, in which meiotic nondisjunction of X chromosomes produces -0.1% XO animals in self-fertilization (Fig. 1b). These XO animals are males and produce 50% nullo-X sperm that give rise to male progeny[13], which increases male ratios in the population[14] (Fig. 1b). This XX/XO sex chromosome system is evolutionarily stable and did not change with the three independent transitions from gonochorism to androdioecy in well-studied *Caenorhabditis*[15]. Here, we show a completely different pattern in the related genus *Pristionchus* that contains the second nematode model organism, *Pristionchus pacificus*. Analysis of 47 *Pristionchus* species revealed rapid chromosome evolution with five distinct karyotypes, four of which are found in the eight androdioecious species.

The model organism *P. pacificus* has well-developed forward and reverse genetic tools and has been intensively studied with regard to the developmental plasticity of its mouth-form feeding structures, associated predation, nervous system wiring and self-recognition mechanisms[16–21]. *C. elegans* and *P. pacificus* both have six chromosomes and share an XO sex chromosome system[22], which was long thought to represent the common sex determination system in nematodes[15,23]. *Pristionchus* nematodes are soil nematodes that are reliably found in association with scarab beetles[24,25] (chafers, stag beetles and dung beetles). Intense worldwide samplings resulted in a collection of 47 culturable *Pristionchus* species with the gonochorist *P. exspectatus* representing the sister species of *P. pacificus*, both of which can still form partially fertile hybrids[10,26]. A previous study indicated that the *C. elegans*-like karyotype of *P. pacificus* resulted from a recent chromosomal fusion event[27]. Specifically, *P. pacificus* and *P. exspectatus* have six chromosomes with an XO system (hereafter 6, XO) and six chromosomes with an XY system (hereafter 6, XY), respectively; while the outgroup gonochoristic species *P. occultus* has seven chromosomes and an XO sex chromosome system[27] (Fig. 1c; hereafter 7, XO). The chromosome fusion between an autosome and the X chromosome in *P. exspectatus* forms a neo-sex chromosome, which gives rise to the new XY system (Fig. 1c). Thus, among the three most closely related species, two independent chromosomal fusion events occurred although all three species can form partially fertile hybrids.

This unexpectedly rapid karyotype evolution inspired a systematic analysis across the genus *Pristionchus*. Here, we show through karyotype analysis and representative whole genome sequencing coupled with Hi-C analysis that five distinct karyotypes exist in *Pristionchus*. Four different karyotypes are found in the androdioecious species, and three of them have six chromosomes but no XO system. More detailed genomic analysis in two of these species, *P. mayeri* and *P. entomophagus*, indicated that there are no detectable differences between hermaphrodites and males. Indeed, further experiments revealed the presence of stochastic sex determination (SSD) that is influenced by temperature and is evolutionarily derived from gonochoristic ancestors with an XY chromosome system. Thus, the genus *Pristionchus* exhibits frequent evolution of androdioecy associated with pervasive karyotype evolution and the evolution of different sex determination mechanisms.

## Results

### Rapid karyotype evolution and various sex chromosome systems in *Pristionchus* nematodes

We systematically performed karyotype analysis in 47 *Pristionchus* species that are available as living cultures, studying meiotic (Prophase I) and gamete cells in male gonads (Fig. 1d–h, Supplementary Data 1). Surprisingly, we found rapid evolutionary changes resulting in five distinct karyotypes, four different chromosome numbers and at least 12 independent evolutionary transitions throughout the genus *Pristionchus* (Fig. 1h). These transitions were not directly associated with reproductive mode (Fig. 1h, where androdioecious species are shown in bold letters and the symbol, ♀♂), resulting in a highly diverse pattern of chromosome evolution and likely associated sex chromosome systems. Phylogenetic estimates of chromosome evolution revealed five important patterns. First, we found four different haploid chromosome numbers, $n = 5, 6, 7$ or 12 in the different *Pristionchus* species, in contrast to the conserved chromosome number ($n = 6$) in the genus *Caenorhabditis*. Second, the 7, XO karyotype ('$n = 7$, XO' in Fig. 1h) is putatively ancestral and found in all basal species of *Pristionchus* regardless of the mode of reproduction, i.e. androdioecious (*P. fissidentatus*) and gonochoristic (*P. paulseni*) species. Third, at least six of the observed chromosome fusions resulted in a state of six chromosomes without an associated XO system. Specifically, 24 of the 25 *Pristionchus* species with $n = 6$ do not have an XO sex chromosome system ('$n = 6$ w/o XO' in Fig. 1h). Thus, *P. pacificus* is the only species of the genus sharing the 6, XO karyotype with *C. elegans* ('$n = 6$, XO' in Fig. 1h). Fourth, there is one species (*P. neolucani*) with only 5 chromosomes that likely resulted from an autosomal fusion ('$n = 5$ w/o XO' in Fig. 1h). Finally, we found one polyploidization event in *P. maupasi* resulting in 12 chromosomes, a very recent evolutionary event that will be analysed elsewhere ('$n = 12$ w/o XO' in Fig. 1h). Together, these findings document an unexpectedly frequent chromosome evolution and sex chromosome turnover in *Pristionchus*. Species without a XO system might have a XY system instead, as *P. exspectatus*. However, such a scenario cannot fully explain the karyotype of all species, in particular the four androdioecious species that have lost the XO system. In the following, we use genome sequencing and experimental and reverse genetic approaches to provide insight into the sex determination mechanisms and the evolution of androdioecy in *Pristionchus*.

### The ancestral *Pristionchus* 7, XO karyotype corresponds to conserved nematode Nigon elements

To study the putative ancestral 7, XO unfused chromosomal pattern of basal *Pristionchus* species, we generated a chromosome-level de novo genome assembly in the androdioecious species *P. fissidentatus*. We performed genome sequencing using the PacBio HiFi platform and conducted Hi-C analysis to scaffold the assembly, which resulted in seven long chromosomes (Supplementary Fig. 1a). The size of the *P. fissidentatus* genome is 249 Mb and thus substantially larger than the 158 Mb genome of *P. pacificus*. We predicted a total of 29,244 genes in

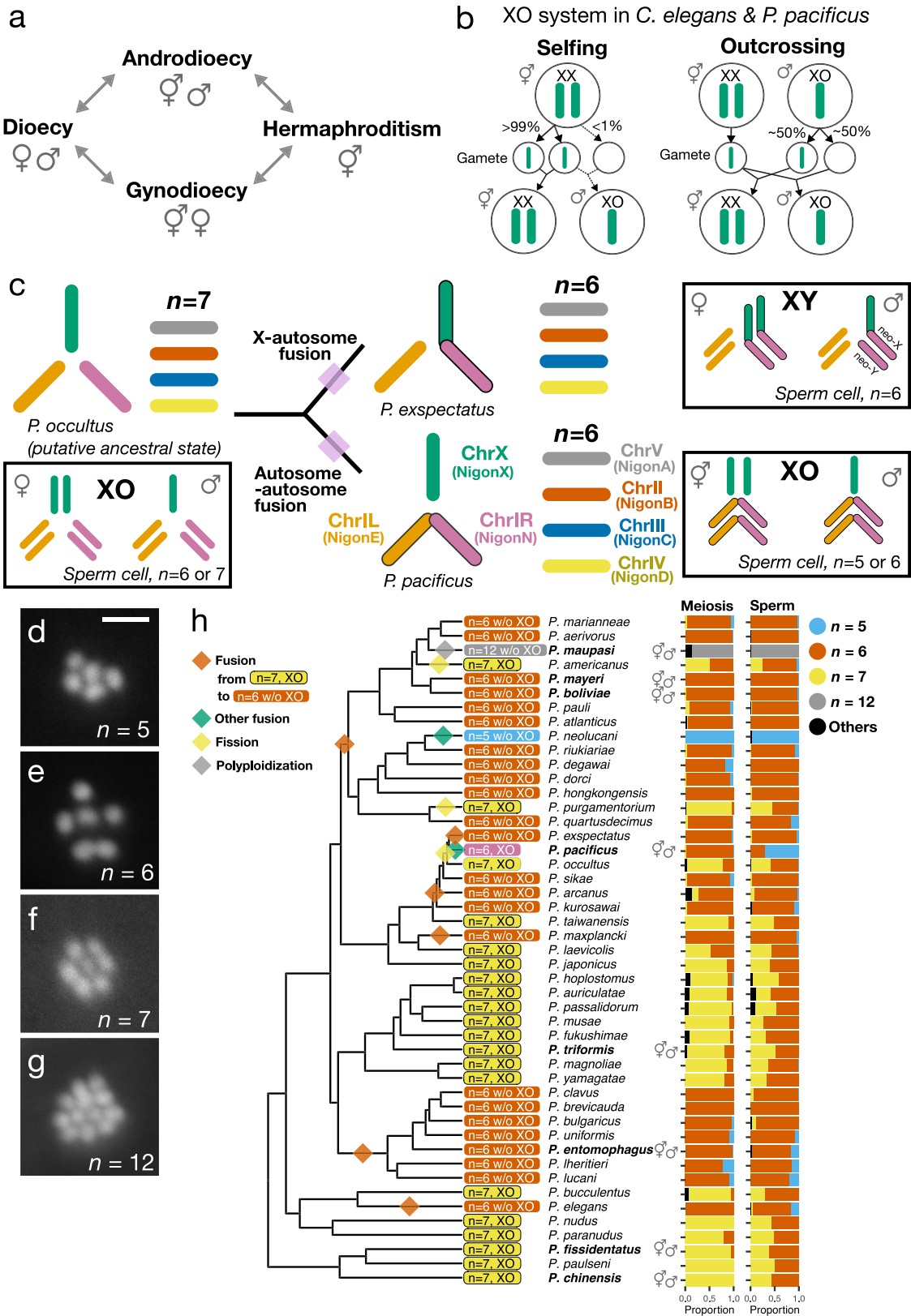

the *P. fissidentatus* genome that contain 89% benchmarking universal single-copy orthologues (BUSCO) as complete single-copy genes (Table 1).

We conducted comparative synteny analysis of the *P. fissidentatus* genome with the high-quality genomes of *P. pacificus* and *P. expectatus*. Based on the genomic positions of the ortholog genes, we found chromosome-level macro-synteny between all three species (Fig. 2a, b

and Supplementary Figs. 3, 4). Specifically, the *P. fissidentatus* genome is organized into seven chromosomes that correspond to the *P. pacificus* chromosomes ChrII, ChrIII, ChrIV, ChrV and ChrX, whereas the left (ChrIL) and the right regions (ChrIR) of *P. pacificus* chromosome I are two separate chromosomes in *P. fissidentatus*. This finding is consistent with the previous observation of independent chromosomal fusions in *P. pacificus* and *P. expectatus*, resulting in their 6, XO and 6,

**Fig. 1 | Chromosome evolution of *Pristionchus* nematodes with independent evolution of androdioecy. a** Proposed evolutionary path between sexual systems after Weeks, 2006[7]. **b** XO system in *C. elegans* and *P. pacificus*. Inheritance of X chromosomes (shown as green bars) and sex determination are visualized in selfing (left) and outcrossing (right). **c** Chromosome evolution between the closely related species *Pristionchus occultus*, *P. expectatus* and *P. pacificus*. Coloured bars indicate the seven conserved chromosomal elements (i.e. Nigon elements). Three chromosome elements, *P. pacificus* ChrI left (NigonE, orange), ChrI right (NigonN, purple) and ChrX (NigonX, green), which are involved in fusion events, are arranged in a tri-radial pattern. Schematic diagrams in boxes represents diploid chromosome number of the three elements in the different sexes. Chromosomes stained by Hoechst 33342 in sperm cells of *P. neolucani* (**d**, $n = 5$), *P. mayeri* (**e**, $n = 6$), *P. fissidentatus* (**f**, $n = 7$), and *P. maupasi* (**g**, $n = 12$). In those species, we tested 259, 204, 86

and 83 cells, in which 252, 199, 33 and 82 cells indicated the same chromosome number, respectively. The scale bar indicates 2 μm. **h.** Karyotype analysis of 47 *Pristionchus* species with phylogenetic estimation. Proportion of male meiotic cells and sperms with each chromosome number is shown in the colored bar graph on the right side of the species name. The sample number is shown in Supplementary Data 1. Karyotypes (left of the species names) were categorized based on those chromosome numbers in male meiosis and sperm, as well as variations of chromosome number in sperm. Species labelled in bold letters are androdioecious, while others are gonochoristic species. The phylogenetic tree is based on a previous phylotranscriptome analysis[9]. The branch length is fitted by chronos function in R package, ape (lambda = 1)[76]. Evolutionary events were parsimoniously estimated and are shown as rhombuses.

XY karyotypes, respectively, both of which are derived from the 7, XO chromosome state as seen in *P. occultus*[27] (Fig. 1c). We also observed intra-chromosomal rearrangements (Supplementary Figs. 3, 4), indicating that rearrangements within sub-regions of chromosomes (i.e. chromosome left-arm, centre and right-arm) occurred frequently, whereas rearrangements between different chromosomes are very rare. Comparative synteny analysis of *P. fissidentatus* with other outgroup nematodes (Supplementary Fig. 2a–d) indicated that the seven chromosomes of *P. fissidentatus* represent the conserved nematode chromosome elements previously defined as 'Nigon elements'[28,29] (Supplementary Fig. 2e). We also confirmed that 97.5% of detected orthogroup genes previously assigned to Nigon elements were located on the corresponding *P. fissidentatus* chromosomes (Supplementary Table 1). The seven *P. fissidentatus* chromosomes, ChrI, ChrII, ChrIII, ChrIV, ChrV, ChrVI and ChrVII, correspond to the Nigon elements NigonE, NigonB, NigonC, NigonD, NigonA, NigonN and NigonX, respectively. Therefore, in the following paragraphs, we use the Nigon-element terminology after synteny was confirmed with *P. fissidentatus*. Taken together, the chromosomal ground state of *Pristionchus* represents seven conserved chromosomal elements that are also highly conserved across nematodes. This ancestral 7, XO karyotype underwent independent chromosomal fusions in multiple independent *Pristionchus* lineages.

### Independent chromosome fusions of the same three chromosome elements in *Pristionchus* nematodes

The major karyotype transition in *Pristionchus* (Fig. 1h) resulted in the $n = 6$ karyotype including several androdioecious species. To identify the genomic basis of these transitions, we performed PacBio Hi-Fi whole-genome sequencing and conducted Hi-C analysis of chromosome-level genome assemblies in two androdioecious species, *P. mayeri* and *P. entomophagus*. The genome size of *P. mayeri* and *P. entomophagus* are 291 Mb and 249 Mb, with a prediction of 33,526 and 30,877 genes and BUSCO completeness of 87% and 82%, respectively (Table 1). Both genome assemblies are organized into six

chromosomes and comparative synteny analysis with the *P. fissidentatus* genome revealed chromosome fusions involving different chromosome elements. Specifically, NigonE and NigonX are fused in *P. mayeri* (Fig. 2c and Supplementary Fig. 5), whereas NigonN and NigonX are fused in *P. entomophagus* (Fig. 2d and Supplementary Fig. 6). Note that the latter represents the same fusion pattern as previously seen in *P. exspectatus* (Fig. 2b). Thus, *P. mayeri*, *P. entomophagus*, *P. exspectatus* and *P. pacificus* all show chromosomal fusions involving the same three conserved elements, namely NigonE, NigonN and NigonX. However, these species exhibit all three possible chromosomal fusions scenarios, i.e. NigonE-NigonX in *P. mayeri*, NigonN-NigonX in *P. entomophagus* and *P. exspectatus* and NigonE-NigonN in *P. pacificus* (Fig. 2e–h). Thus, *Pristionchus* nematodes exhibit independent chromosomal fusions of the same three conserved elements, NigonE, NigonN and NigonX, throughout speciation (Fig. 2).

Gene density analysis revealed that the different chromosomal fusion events have affected chromosome structures in distinct ways. Whole-chromosome rearrangements were observed in the NigonX regions of *P. mayeri* and *P. entomophagus*, resulting in high gene density regions near the centres of the fused chromosomes (indicated by heat panels along the axes of Fig. 2g, h). In contrast, intra-chromosomal rearrangements, as seen between *P. fissidentatus* and *P. pacificus* or *P. fissidentatus* and *P. exspectatus*, are limited to sub-regions of chromosomes. In these cases, fused chromosomes retain the ancestral positions of high gene density regions (i.e. centres of the ancestral chromosomes) (Fig. 2e, f). These findings together with the phylogenetic pattern of karyotype evolution suggest that the fusion events observed in the *P. mayeri* and *P. entomophagus* lineages occurred in ancestral branches, compared to the recent fusions in *P. pacificus* and *P. exspectatus* (Fig. 1h). Furthermore, these findings indicate that regions with high gene density on chromosomes are initially unaffected by chromosomal fusions; but over long evolutionary timescales, these regions will rearrange from the ancestral centres of density to new regions in the centres of fused chromosomal structures. Therefore, chromosome fusions might play a dominant role in gene positioning and repatterning over long evolutionary time scales.

### No genomic differences between sexes in *P. mayeri*

The majority of *Pristionchus* species have six chromosomes, but only *P. pacificus* exhibits the *C. elegans*-type 6, XO karyotype (Fig. 1h). In contrast, all the other 24 *Pristionchus* species with six chromosomes do not form nullo-X sperm (Fig. 1h). To obtain insight into potential sex-linked genomic regions of these species, we first focused on the androdioecious *P. mayeri* because this species forms regular males and multiple strains from different geographic locations are available[30]. To determine if the genomes of androdioecious hermaphrodites and males of *P. mayeri* are really identical, we re-sequenced the genomes of both sexes separately and mapped these reads back to the *P. mayeri* assembly to identify potential sex differences in coverage. Indeed, depth of coverage analysis showed similar density distributions in all chromosomes for both sexes (Fig. 3a). As control, we sequenced

### Table 1 | Chromosome-level genome assembly of the three androdioecious species

| Species | *P. fissidentatus* | *P. mayeri* | *P. entomophagus* |
|---|---|---|---|
| Strain | RS5133 | RS5460 | RS0144 |
| Chromosome No. | 7 | 6 | 6 |
| Genome size [Mb] | 249.0 | 290.8 | 244.9 |
| Scaffold N50 [Mb] | 36.3 | 48.8 | 38.6 |
| No. Gene | 29, 244 | 33, 526 | 30, 877 |
| BUSCO statistics[a] | S: 89%, D: 3.7%, F: 5.2%, M: 2.4% | S: 86.9%, D: 5.3%, F: 3.8%, M: 4.0% | S: 81.5%, D: 4.8%, F: 3.6%, M: 10.1% |

*S* complete single copy, *D* complete duplicated, *F* fragmented, *M* missing.
[a]BUSCO analysis using BUSCO ver. 3.1.0 with nematode_odb9 ($N = 982$).

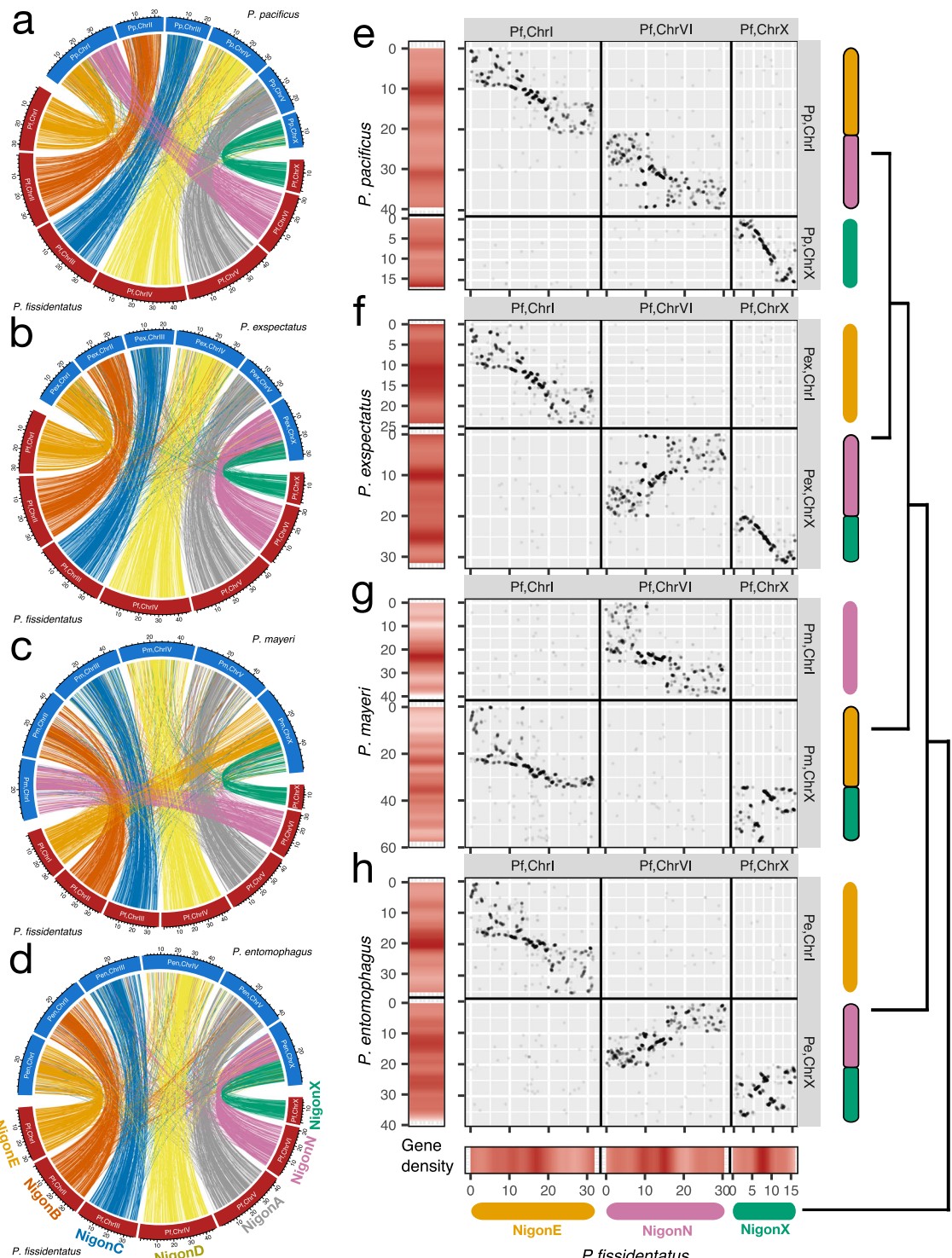

**Fig. 2 | Comparative synteny analysis using chromosome-level assemblies of androdioecious *Pristionchus* nematodes.** Circos plot of comparative synteny between the basal species, *P. fissidentatus* and *P. pacificus* (**a**); *P. expectatus* (**b**); *P. mayeri* (**c**) and *P. entomophagus* (**d**), respectively. For each species pair, three thousand pairs of one-to-one orthologous genes were randomly selected, and their genomic positions were displayed as links. Dot plot of comparative synteny between *P. fissidentatus* and *P. pacificus* (**e**); *P. expectatus* (**f**); *P. mayeri* (**g**) and *P. entomophagus* (**h**), respectively, for the chromosomes involved in chromosome fusions. Red heat panels in the axes indicate trends of gene density (i.e. LOESS regression curves) as shown in Supplementary Fig. 7. Fusion patterns are represented by coloured bars similar to Fig. 1. The species phylogeny is displayed on the right.

androdioecious hermaphrodites and males of *P. pacificus* and females and males of *P. expectatus* with their 6, XO and 6, XY karyotypes, respectively. Here, we observed a $\log_2$ coverage difference of 1 between sexes in the ancestral X chromosome regions of 6, XO (Figs. 3b, 4), XY (Fig. 3c). These findings indicate that *P. mayeri* has no

coverage difference in the X chromosomes between androdioecious hermaphrodites and males and a karyotype of $n = 6$ in both sexes with two ancestral X chromosomes (hereafter, 6, XX). Thus, *P. mayeri* sexes have identical genomes without distinct sex chromosomes or larger sex-linked genomic regions.

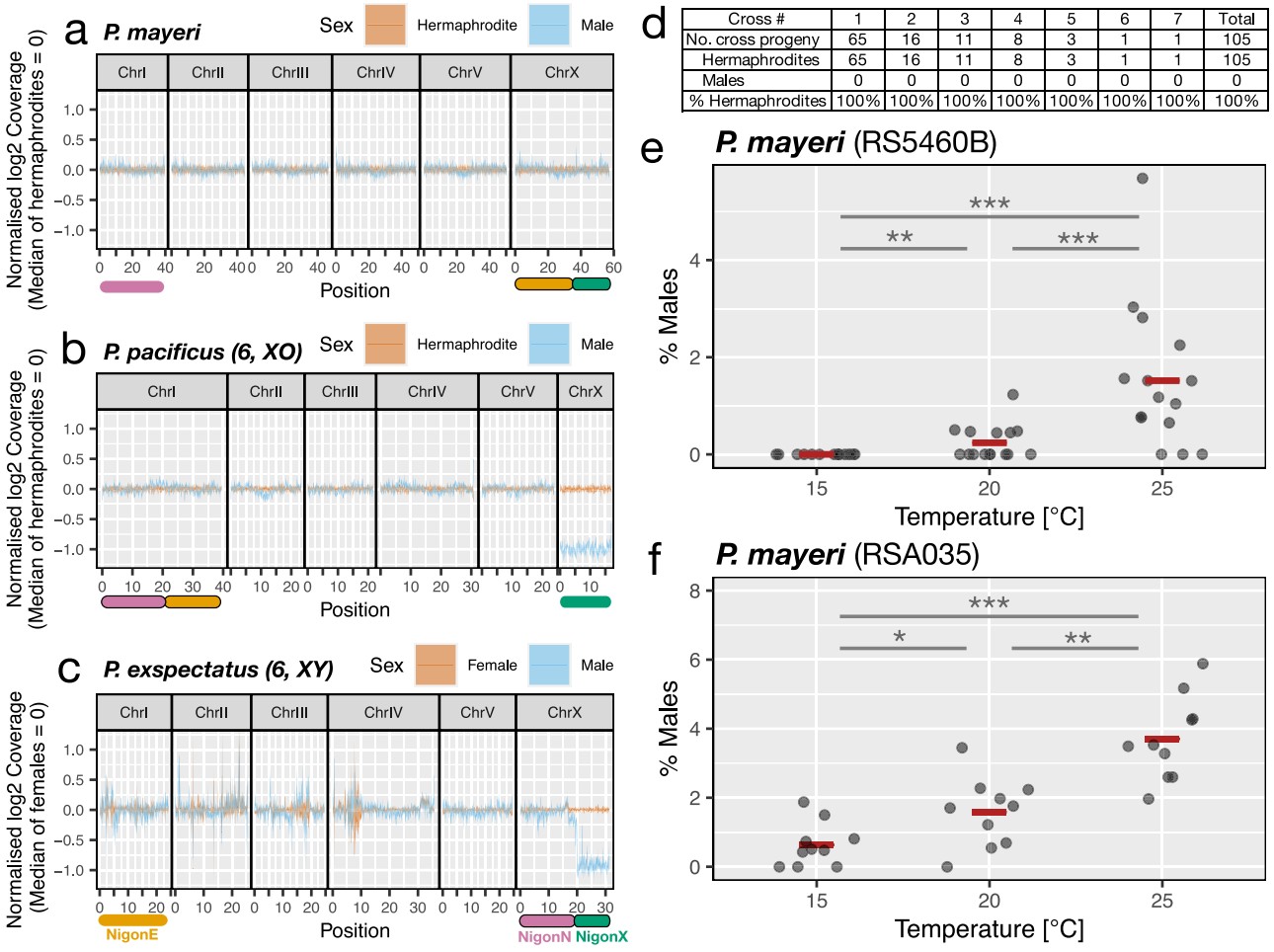

**Fig. 3 | Thermosensitive stochastic sex determination of *P. mayeri*.** Comparison of sequence read coverage depth between the sexes of *P. mayeri* (**a**), *P. pacificus* (**b**) and *P. exspectatus* (**c**), respectively. Average log$_2$ coverage of 100-kb sliding windows is normalised for each individual and median of the normalised coverage were presented as a line. The area between the first and third quartiles were shown in the light-colour ribbon. Colored bars on the axes represent chromosome elements corresponding to NigonE (orange), NigonN (purple) and NigonX (green). *P. pacificus* and *P. exspectatus* have 6, XO and 6, XY karyotype, respectively, and exhibit log$_2$ coverage difference specifically in the ancestral X chromosome regions (i.e. NigonX). **d** The number of hermaphrodite and male progeny in crosses between different strains of *P. mayeri*. Only 7 out of 10 crossing experiments yielded cross progeny. Test of temperature effects on the percentage of male production in the *P. mayeri* strains RS5460B (**e**, $N = 15$ each) and RSA035 (**f**, $N = 10$ each). Each jitter point represents the male ratio of a single biological replicate with the

red bar indicating the average across replications. Statistic tests were conducted with GLMM for pairwise comparison. For RS5460B, the effects of temperature differences on the log odds (and their 95% confidence intervals) were estimated as 21.69 ([20.9, 33.5]), 1.82 ([1.09, 2.92]), and 24.1 ([23.6, 35.4]) for 15 °C *vs.* 20 °C, 20 °C *vs.* 25 °C, and 15 °C *vs.* 25 °C, respectively. Those $\chi^2$ statistics are 8.80, 15.3 and 33.4, respectively. Those Bonferroni-adjusted *p*-values are $9.03 \times 10^{-3}$, $2.78 \times 10^{-4}$ and $2.24 \times 10^{-8}$, respectively. For RSA035, the effects of temperature differences on the odds ratio (and their 95% confidence intervals) are estimated as 0.966 ([0.326, 1.77]), 0.852 ([0.302, 1.45]), and 1.81 ([1.18, 2.63]) for 15 °C *vs.* 20 °C, 20 °C *vs.* 25 °C, and 15 °C *vs.* 25 °C, respectively. Those $\chi^2$ statistics are 6.68, 8.93 and 23.4, respectively. Those Bonferroni-adjusted *p*-values are $2.92 \times 10^{-2}$, $8.39 \times 10^{-3}$ and $3.98 \times 10^{-6}$, respectively. The degrees of freedom of all tests are 1. The Bonferroni-adjusted *p*-values are one-sided due to goodness-of-fit test and represented as stars (*$p < 0.05$, **$p < 0.01$, ***$p < 0.001$).

## Cross progeny of *P. mayeri* hermaphrodites and males are all hermaphrodites

In theory, sex determination in *P. mayeri* with its 6, XX karyotype in androdioecious hermaphrodites and males, could be i) genetic (GSD), with a single sex determination locus or a small genomic region that goes undetected in genome sequencing, ii) environmental (ESD), or iii) stochastic (SSD), as recently suggested for the plant parasitic nematode *Bursaphelenchus okinawaensis*[11]. However, GSD is unlikely in *P. mayeri* because self-fertilizing hermaphrodites form spontaneous males with low frequency, similar to *P. pacificus* and *C. elegans*[30]. To formally distinguish between these three possibilities, we performed mating experiments between males and old androdioecious hermaphrodites that had run out of self-sperm, a method that is regularly used in *P. pacificus*[27]. We worked with two natural isolates of *P. mayeri*, RS5460 and RSA035 from La Réunion Island. These two strains display a single-nucleotide-polymorphism (SNP) in the SSU ribosomal spacer

allowing the distinction of cross progeny. In case of GSD, about half of the progeny of a cross should be males, such as in *P. pacificus* and *C. elegans*. In contrast, if sex determination is environmental or stochastic, outcrossing and selfing might result in similar sex ratios with a minority of males being formed.

The two *P. mayeri* isolates RS5460 and RSA035 form less than 1% males under standard laboratory conditions at 20 °C. From 10 crosses of single RS5460 males with single, old RSA035 androdioecious hermaphrodites, we observed a total of 150 offspring all of which were hermaphrodites. We sequenced all 150 animals and found that 105 hermaphrodites were indeed heterozygous at the site of the SSU polymorphism indicating that they are cross progeny (Fig. 3d). In contrast, 44 hermaphrodites were homozygous for the RSA035 SNP indicating that they result from self-fertilization, whereas the remaining hermaphrodite ($N = 1$) had an undetermined SNP pattern. These findings indicate that cross progeny between

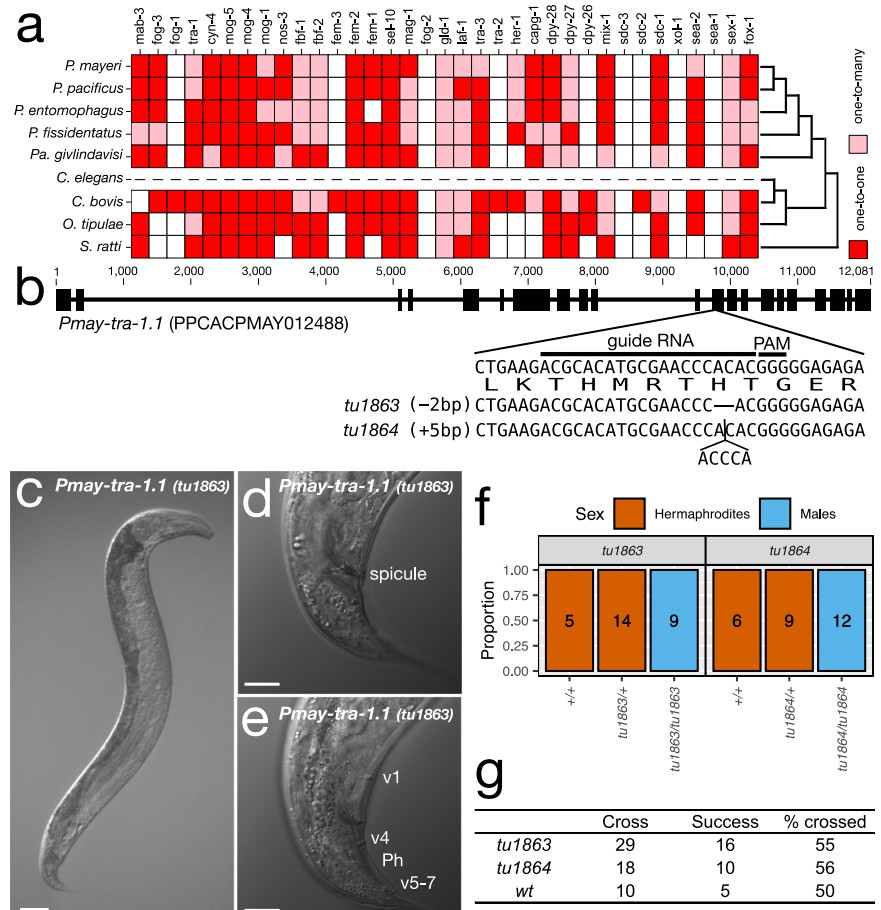

**Fig. 4 | The conserved transcription factor TRA-1 mediates environmental sex determination. a** Orthologs of *C. elegans* genes involved in sex determination in *Pristionchus* and outgroup nematodes. The colour panels indicate orthology relationship. Genes positioned more to the right are upstream in the pathway. The species phylogeny is presented on the right. **b** Establishment of CRISPR engineering in a nematode system, *P. mayeri*. The genomic Region of *Pmay-tra-1.1* is displayed. Exons are represented by black boxes. A guide RNA was designed within the conserved zinc-finger domain. Two mutants with 2-bp deletion (*tu1863*) and 5-bp insertion (*tu1864*) were isolated. Phenotypes of homozygous *Pmay-tra-1.1* mutants in whole body (**c**) and posterior male-specific organs (**d**, **e**). Scale bars indicate 50 μm (**c**) and 20 μm (**d**, **e**). The present individual was one of ten tested male-like individuals, all of which indicated the same morphology. **f** Proportion of different sexes in *Pmay-tra-1.1* mutants. The number displayed in the centre of each column represents sample size. **g** Percentage of successful crosses using *Pmay-tra-1.1* mutant males.

androdioecious hermaphrodites and males in *P. mayeri* are exclusively of the hermaphroditic sex. Thus, there is no GSD in *P. mayeri* and likely environmental or stochastic factors determine the sex of these worms.

## *P. mayeri* exhibits stochastic sex determination influenced by temperature

Next, we wanted to know if the formation of males in *P. mayeri* can be induced by environmental stimuli and how it relates to ESD or SSD. Typically, ESD refers to systems where an environmental factor causes animals to become male or female, such as in some reptiles[31,32]. In SSD, stochastic gene expression of a bistable molecular pathway determines the sex of an individual and might additionally be influenced by environmental factors[33]. It has been argued that SSD is often disregarded and that examples of SSD are subordinated as ESD[33]. Temperature is one of the fundamental environmental factors involved in sex determination in diverse taxa[34,35]. Therefore, we tested different temperatures to induce males in *P. mayeri*. We found that male ratio in the culture was associated with culture temperature in *P. mayeri* RS5460. For example, worm cultures grown at 20 °C showed about 0.24% male formation, whereas increasing culture temperature to 25 °C resulted in an average of 1.46% males ($N = 15$ each, GLMM test between 20 °C *vs*. 25 °C, Bonferroni-adjusted $p = 2.7 \times 10^{-4}$; Fig. 3e). In

contrast, cultures reared at 15 °C produced no males in the following generation ($N = 15$ each, GLMM test between 15 °C *vs*. 25 °C, Bonferroni-adjusted $p = 0.0090$; Fig. 3e). Similarly, we also found an increase in male formation in the other strain of *P. mayeri* (Fig. 3f). Thus, high temperature increased the male ratio of *P. mayeri* but individual sexes within each culture are stochastically determined despite having the same genotype and environmental conditions. Therefore, we concluded that *P. mayeri* exhibits SSD with the induction of males at elevated temperatures.

## The terminal sex determination regulator TRA-1 is involved in SSD in *P. mayeri*

Next, we asked whether SSD in *P. mayeri* still relies on conserved sex determination regulators as originally defined in *C. elegans*. For that, we first identified the *P. mayeri* orthologs of the major *C. elegans* sex determination genes using OrthoFinder with newly annotated genes of *Pristionchus* and other outgroup species (Fig. 4a; Supplementary Data 2). We found that all downstream sex determination regulators that control somatic sexual differentiation are conserved in *P. mayeri* (left genes in Fig. 4a, except for *fog-1* and *fem-3*). This includes the terminal regulator *tra-1*, which encodes a transcription factor acting downstream in the sex determination gene regulatory network[13] (Fig. 4b). In contrast, upstream factors such as the primary switch

genes *xol-1* and *sea-1* were not found, suggesting that these genes are either rapidly evolving or even totally absent. Note that independent gene birth and death events in different nematode lineages have resulted in 'one-to-one' or alternatively, 'one-to-many' or 'many-to-many' gene copy relationships between different species (Fig. 4a).

To study if SSD in *P. mayeri* still involves part of the sex determination machinery originally identified in *C. elegans*, we selected the transcription factor TRA-1 as a candidate locus. *P. mayeri* has two copies of the *tra-1* gene (PPCACPMAY012488 and PPCACPMAY003417, which are designated as *Pmay-tra-1.1* and *Pmay-tra-1.2*, respectively). Note that *Pmay-tra-1.1* has a longer region with high sequence similarity to *Cel-tra-1* (319 a.a. in *Pmay-tra-1.1 vs.* only 121 a.a. in *Pmay-tra-1.2*) and therefore, we used *Pmay-tra-1.1* to perform reverse genetics using the CRISPR/Cas9 technology (Fig. 4b). This technology works robustly in *P. pacificus*[36] and a distant relative *Allodiplogaster sudhausi*[37,38]. Indeed, we were able to generate CRISPR/Cas9-induced gene knockouts using the same protocol as for *P. pacificus*. We obtained three alleles of *Pmay-tra-1.1(tu1863, tu1864, tu1865)* with independent frameshift mutations, all resulting in premature stop codons. Surprisingly, we observed a fully penetrant morphological male phenotype in homozygous mutants (Fig. 4c–f). These findings are different from *P. pacificus*, where *tra-1* mutants do not result in fully penetrant male phenotypes[22]. To test if *Pmay-tra-1.1* mutant males are functional, we crossed mutant animals with old wild type hermaphrodites that had run out of sperm. Indeed, we found that roughly half of the mutant males produced cross progeny (Fig. 4g). Thus, mutations in *Pmay-tra-1.1* result in a complete *transformer* phenotype of XX animals into males that can generate cross progeny with androdioicious hermaphrodites. Therefore, *Pmay-tra-1.1* acts as a sex determination switch gene. In conclusion, these experiments suggest that SSD in *P. mayeri* relies at least in parts on a conserved sex determination machinery similar to *C. elegans*.

## SSD is derived from an ancestral XY sex chromosome system
Next, we wanted to study the origin of the *P. mayeri* SSD system and its relationship to any GSD system. Of the 24 *Pristionchus* species with six chromosomes but no nullo-X sperm, 20 are gonochoristic species that form males and females with similar ratios. In such species, the karyotype could potentially be 6, XX with SSD. Alternatively, such species might have a sex determination system that relies on autosomal sex-determining loci, such as in some populations of the housefly *Musca domestica*[39,40]. A third possibility would be that they exhibit a 6, XY karyotype with conserved sex-linked genomic regions, such as in *P. exspectatus* with its young neo-Y chromosome that shows few genomic differences to the neo-X chromosome[27]. To study the ancestral karyotype in species related to *P. mayeri*, we investigated two gonochoristic relatives. First, we studied *Pristionchus atlanticus*, which has a more basal position in the same sub-clade of the genus, and second, we investigated *Pristionchus aerivorus* that is more closely related to *P. mayeri* (Fig. 1h). For this, we developed a reference-free method for coverage analysis of short-read genome sequences, utilizing conserved chromosome-specific 21-mers shared between *P. pacificus* and *P. fissidentatus*. In this methodology, we separately sequenced whole genomes from single worms of different sexes and identified differences in coverage depth of specific chromosomes based on the count of the conserved 21-mers within the reads (Fig. 5a). As controls, we also examined the genomes of individual worms of *P. fissidentatus* (7, XO), *P. pacificus* (6, XO), *P. exspectatus* (6, XY) and *P. mayeri* (6, XX).

We found no striking coverage difference across chromosome regions for all species, except for regions of the X chromosome. Specifically, regions of the genome that correspond to the ancestral X chromosome show different coverage depths between sexes in *P. fissidentatus*, *P. pacificus*, *P. exspectatus*, and

*P. atlanticus* (Wilcoxon-Mann-Whitney test of 'NigonX' vs. 'Others', Bonferroni-adjusted $p = 7.80 \times 10^{-8}$, $1.12 \times 10^{-4}$, $1.12 \times 10^{-4}$, and $7.80 \times 10^{-8}$, respectively, Fig. 5a), but not in *P. aerivorus* and *P. mayeri* (Bonferroni-adjusted $p = 1$ and 1, respectively; Fig. 5a). These results suggest that *P. atlanticus* has a 6, XY karyotype similar to *P. exspectatus*, whereas *P. aerivorus* has a 6, XX karyotype. Therefore, the 6, XX karyotype of *P. mayeri* is likely derived from a 6, XY karyotype (Fig. 5b). This observation is also consistent with the NigonE-NigonX fusion in the *P. mayeri* genome (Fig. 2b). In summary, these results indicate that the sex chromosome evolution predated the evolution of the primary sex determination mechanisms in *P. mayeri*.

## SSD has independently evolved in a second *Pristionchus* lineage
Finally, we wanted to know if SSD is common and independently evolved in different *Pristionchus* lineages, or alternatively, represents an oddity of *P. mayeri*. Therefore, we investigated another androdioecious species of the genus. The European species *P. entomophagus* is only distantly related to *P. mayeri* but shares the same $n = 6$ karyotype without an XO system (Fig. 1h). In addition, *P. entomophagus* is also not closely related to *P. pacificus* and as such, represents another major lineage of the *Pristionchus* phylogeny[10,41]. Comparison of the coverage depth of whole genome sequence reads mapped to the *P. entomophagus* reference genome between sexes indicated that there is no difference between androdioecious hermaphrodites and males across the entire genome (Fig. 6a). Therefore, the *P. entomophagus* karyotype is also 6, XX, similar to what we found in *P. mayeri*.

To determine if *P. entomophagus* also relies on SSD we performed similar experiments as described above for *P. mayeri* (Fig. 6b). Under standard laboratory conditions at 20 °C, *P. entomophagus* forms males with low frequency (0,5%) (Fig. 6b). However, when we cultured the same strain at 15 °C, we observed a nearly three-time increase in male formation to 1,3% (GLMM test, $p = 0.0020$ Fig. 6b). Thus, *P. entomophagus* and *P. mayeri* independently evolved SSD that relies on temperature; however, with different directionality. In *P. mayeri*, male induction is seen at high temperatures, whereas in *P. entomophagus* it is observed at low temperature. Similarly, *P. entomophagus* and *P. mayeri* have independently evolved a 6, XX karyotype without genomic differences between androdioecious hermaphrodites and males (Fig. 6a). Taken together, our analysis of multiple androdioecious *Pristionchus* species indicates pervasive karyotype and sex chromosome evolution, some of which are coupled with the evolution of sex determination mechanisms.

## Discussion
Here, we show frequent chromosome evolution and sex chromosome turnover in the nematode genus *Pristionchus* and the independent acquisition of stochastic sex determination influenced by temperature in two independently evolved androdioecious species. Both of these species are derived from gonochoristic ancestors with GSD mechanisms. These findings build on dense taxon sampling of *Pristionchus*, which was facilitated by the strong association of these nematodes with scarab beetles worldwide[10]. As a result, nearly 50 *Pristionchus* species are available as living cultures with a robust transcriptome-based phylogeny that allows proper mapping of karyotype and genome evolution[9,10]. We obtained two conclusions with general implications.

First, rapid karyotype evolution in *Pristionchus* involved multiple independent chromosome fusion events from an ancestral 7, X0 karyotype. This finding is compatible with evolutionary mechanisms of reproductive isolation by chromosome evolution[27]. Indeed, chromosome evolution was long known to facilitate speciation in mammals, reptiles and butterflies[42–46], whereas studies in nematodes suggested relative karyotype stability[28,29,47]. However, our findings indicate that different nematode taxa might involve distinct speciation

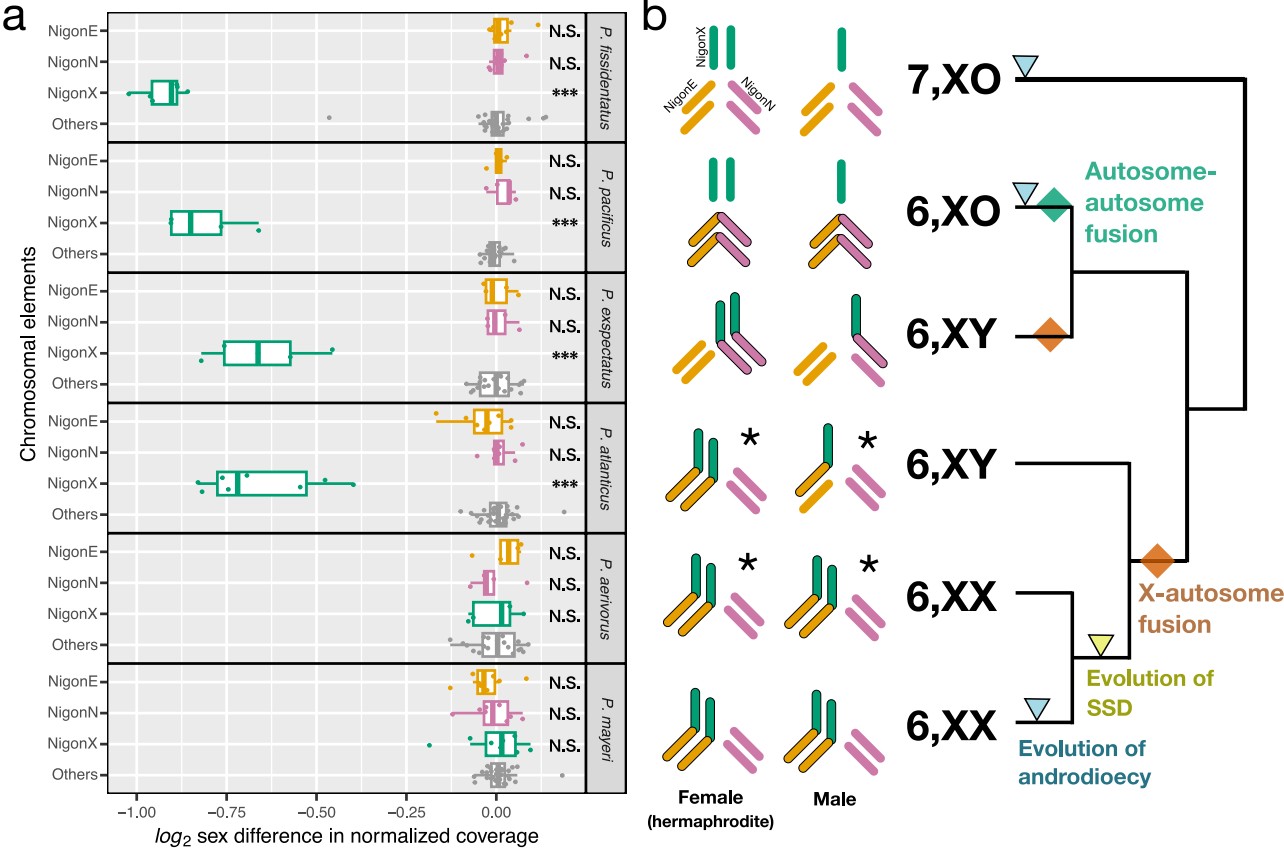

**Fig. 5 | Putative evolutionary history of sex determination in *Pristionchus*.**
**a** Difference in coverage depth of chromosome-specific 21-mers between sexes. The category labelled as 'Others' contains data from all other chromosomal elements (i.e. NigonA, NigonB, NigonC and NigonD). All biological replicates are shown as jitter plots (N = 8 for *P. fissidentatus, P. atlanticus* and *P. mayeri*; N = 5 for *P. pacificus, P. exspectatus, P. aerivorus*) while the box plot shows statistics (upper whisker, the largest data point less than the third quartile +1.5 × interquartile range; upper bound, the third quartile; centre line, median; lower bound, the first quartile; lower whisker, the smallest data point more than the first quartile −1.5 × interquartile range). The average $log_2$ coverage depth of each chromosome for each species is normalized with the autosomal average and compared between randomly paired male and female (or hermaphrodite) individuals. In each species, the

$log_2$ sex differences of three Nigons were compared with that of 'Others' and tested with Wilcoxon–Mann–Whitney test (W = 102, 115, 256, 35, 26, 100, 47, 43, 100, 169, 117, 256, 39, 55, 50, 187, 145, and 115, from top to bottom, two-sided Bonferroni-adjusted $p = 1, 1, 7.80 \times 10^{-8}, 1, 0.337, 1.13 \times 10^{-4}, 0.523, 1, 7.80 \times 10^{-8}, 1, 1, 1, 0.139, 1,$ and 1, from top to bottom. The two-sided Bonferroni-adjusted $p$-values are shown as stars ($p > 0.05$, N.S.; *$p < 0.05$, **$p < 0.01$, ***$p < 0.001$). **b** Phylogenetic estimation of chromosome evolution. Left schematic diagrams with coloured bars that corresponds to Nigon elements (NigonE, orange; NigonN, pink; and NigonX, green) represent diploid chromosome patterns of each sex. Chromosome patterns of *P. atlanticus* and *P. aerivorus* (marked with star '*') were inferred using karyotyping (Fig. 1h) and coverage data (**a**). Evolutionary events were determined parsimoniously.

mechanisms. For example, comparative studies in *Caenorhabditis* revealed a conservation of the 6, XO karyotype throughout the genus[48–50]. Thus, speciation in *Caenorhabditis* does not involve chromosome fusion events. In contrast, the rapid karyotype evolution as observed in *Pristionchus* might facilitate speciation. Indeed, our case study of closely related species confirmed the effect of chromosome fusions on the evolution of reproductive isolation[27]. These strong differences between *Pristionchus* and *Caenorhabditis* should encourage similar studies in other nematode taxa. Indeed, recent studies suggest that some nematode genera have similar chromosome variation as observed in *Pristionchus*[51,52]. It is important to note that the chromosome fusions observed in *Pristionchus* always involved the same three chromosomal elements, NigonE, NigonN and NigonX. This unexpected finding points towards genomic constraints, the molecular nature of which remain currently elusive. Given that all three possible combinations of fusions have indeed been observed (NigonE-NigonN, NigonE-NigonX, NigonN-NigonX), it will be interesting to see which chromosomes have fused in *P. neolucani*, the only species of the genus with five chromosomes (Fig. 1h). One possibility would be a fusion of NigonE-NigonN-NigonX to become a single chromosome, a hypothesis

that can be tested through genome sequencing. Finally, it is worth to note that our parsimonious estimate of karyotype transitions suggested 12 'state' transitions, but this number might be an underestimate because parallel fusion or fission events might have happened. Nevertheless, this estimate is consistent with the results of genome sequence analysis across *Pristionchus* species. The putative ancestral karyotype (7, XO) in *P. fissidentatus* corresponds to the ancestral chromosomes previously proposed for nematodes as a group[29], followed by independent fusion events in different *Pristionchus* lineages. Once more chromosome-level genome assemblies of *Pristionchus* will be available, future studies may be able to determine how frequently the similar fusion events happened.

Second, we demonstrate SSD that is influenced by temperature in two independent lineages of *Pristionchus*. The parallel evolution of SSD in *Pristionchus* adds to recent explorations of non-model nematodes that has unveiled unprecedented diversity of sex determination. For example, *Bursaphelenchus* nematodes have stochastic sex determination that responds on temperature as *Pristionchus*[11]. In *Auanema* species, which exhibit three sexes (i.e. males, females and hermaphrodites), environmental cues and maternal age alter the ratio

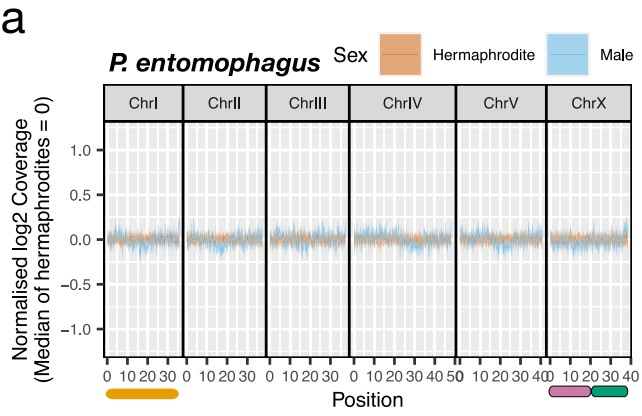

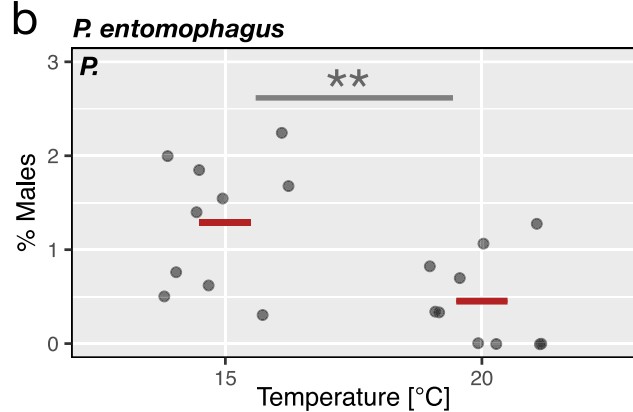

**Fig. 6 | Independent evolution of stochastic sex determination in *P. entomophagus*. a** No coverage difference across the entire genome was observed between sexes in *P. entomophagus*. The average $\log_2$ coverage of 100-kb sliding windows was normalised for each individual and the median of the normalised coverage are presented as a line. The area between the first and third quartiles is shown in the light-coloured ribbon. Coloured bars on the axes corresponds to NigonE (orange), NigonN (purple) and NigonX (green). **b** Test of temperature effects on the percentage of males in *P. entomophagus* ($N = 10$ each). Each jitter point represents the male ratio of a single biological replicate, with the red bar indicating the average across replications. A statistic test was conducted with GLMM for pairwise comparison. The effect of temperature differences on the log odd (and its 95% confidence interval) was estimated as $-1.07$ ([$-1.85$, $-0.462$]). The $\chi^2$ statistics and $p$-value are 9.59 and $1.96 \times 10^{-3}$, respectively. The degree of freedom is 1. The $p$-value is one-sided due to goodness-of-fit test and represented as stars (*$p < 0.05$, **$p < 0.01$, ***$p < 0.001$).

of females to hermaphrodites in XX animals[53,54]. Mitotic chromosome diminution of sex chromosomal regions in *Strongyloides* species spontaneously produces males[55,56]. Thus, evidence in multiple nematode lineages suggest that SSD which is influenced by various environmental factors is much more common than previously assumed. Our findings on the independent evolution of SSD are consistent with the rapid evolution of the upstream sex determination pathway as originally described in the genus *Caenorhabditis*. For instance, *fog-2/gld-1*, which are upstream genes regulating *tra-2*, plays a crucial role in spermatogenesis of *C. elegans* hermaphrodites[57,58] but this mechanism is unique to this species[59,60], and *C. briggsae* exhibits another specific gene, *she-1*, regulating *tra-2* for its hermaphroditism[61]. Additionally, the upstream genes, *xol-1* and *fem-3*, likely experienced rapid protein evolution, which might partially explain why we could not detect some of the genes in species only distantly related to *C. elegans*[62,63]. These findings suggest that sex-determination pathways can evolve rapidly, and that in some cases this results in the decoupling from regulation by sex chromosome number. For SSD, it was proposed that any random fluctuation in the expression of genes at the top of the sex-determination cascade has the potential to drive individual development towards one of the two stable states[33]. In *Pristionchus*, we speculate that stochastic processes in transcriptional, translational and/or post-translational regulation of *tra-1* may play a role in SSD. Similarly, the role of *tra-1* in SSD has also been supported by forward genetic screening of *Bursaphelenchus* although mutant animals exhibited an intersexual phenotype[11] rather than a complete sex reversal as in *Pmay-tra-1.1*. However, in principle, any protein in a signalling pathway might become temperature-sensitive and result in the thermosensitive SSD as seen in these species. Given the availability of genetic technology, including CRISPR/Cas9 system, *Pristionchus* can serve as a potential study system to uncover the molecular mechanism of SSD.

SSD as observed in *P. mayeri* is secondarily derived from an ancestral GSD system that underwent sex chromosome turnover by chromosome fusions. We found genomic evidence for a 6, XY karyotype in *P. atlanticus* (Fig. 5a), which is consistent with the presence of an X-autosome fusion in *P. mayeri* genome (Fig. 2c). The finding of such differences among closely related nematodes can be used to establish first hypotheses about the evolution of androdioecy in nematodes (Fig. 7). In an ancestral dioecious species with a 7, XO karyotype

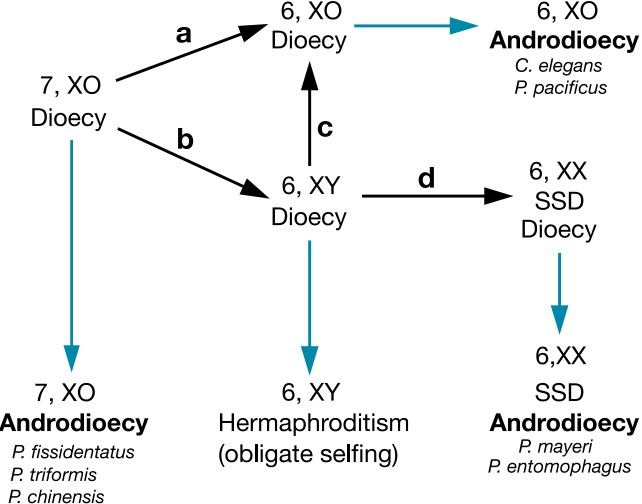

**Fig. 7 | Model of karyotype evolution and evolution of androdioecy.** A model for the evolutionary transitions of karyotypes is represented as arrows labelled with letters: **a** autosome–autosome fusion; **b** X-autosome fusion; **c** loss of a Y chromosome; **d** evolution of stochastic sex determination. Blue arrows represent evolution of hermaphrodites. The indicated species independently evolved androdioecy.

autosome–autosome or X-autosome fusions can result in 6, XO or 6, XY karyotypes, respectively (transitions a and b in Fig. 7). In such a hypothetical case, although sex is determined by counting ancestral X chromosome regions, the Y chromosome is necessary for males in a resulting 6, XY karyotypes, because otherwise, the Y chromosome will be lost subsequently (transition c in Fig. 7). The loss of the Y chromosome would happen if dosage compensation mechanisms extend to the neo-sex chromosome region. However, we did not observe any case of a transition from a 6, XY to a 6, XO karyotype in *Pristionchus*. In contrast, the only 6, XO karyotype found in the genus results from an autosome–autosome fusion in *P. pacificus* (transition a in Fig. 7). If hermaphroditism spontaneously evolved from female individuals through a genetic mutation within a 6, XY karyotype (XX females and XY males), such XX hermaphrodites could not produce XY males

anymore. Instead, they could only produce hermaphroditic offspring through selfing. Such a scenario would lead to a state of strict hermaphroditism with obligate selfing. However, such a transition was also not seen in *Pristionchus*. Instead, *Pristionchus* hermaphrodites evolved in a 6, XX karyotype with SSD (transition d in Fig. 7) as an alternative path to produce males through selfing. Remarkably, this transition has happened at least twice in *Pristionchus* resulting in androdioecy with SSD (Figs. 3 and 6). Our genomic analysis of the closely related dioecious species *P. aerivorus* indicated a 6, XX karyotype (Fig. 5a), suggesting the presence of SSD in a dioecious ancestor. However, we cannot rule out the possibility of an alternative sex determination system in *P. aerivorus*. Together, these findings suggest that the evolution of androdioecy in *Pristionchus* nematodes can occur in a context of variations in sex chromosome and sex determination systems. However, it is important to note that the link between the evolution of SSD and karyotype evolution might be coincidental. Although *P. entomophagus* and *P. mayeri* have indeed evolved SSD from an XX/XY ancestor, whether this is causative remains currently unknown. Importantly, in this study, we did not observe any SSD in species having the ancestral 7 chromosomes. In contrast, two independent evolutionary events to SSD were observed after the sex chromosome–autosome fusions. Interestingly, *Bursaphelenchus* species that gained the SSD also experienced sex chromosome–autosome fusions in their ancestor and have six chromosomes (Supplementary Fig. 2e). However, the total number of examples of SSD in nematodes is currently too small to draw any final conclusions about necessary predispositions of sex chromosomes. Therefore, the evolutionary relationship between sex chromosome evolution and SSD can only be clarified once more SSD cases are documented.

In conclusion, our work adds evolutionary mechanism involved in sex determination to recent findings on the evolution of androdioecy in nematodes. It also suggests that SSD might more frequently evolve than previously assumed. For the genus *Pristionchus*, SSD represents the third example of, in parts, stochastically regulated processes, besides dauer development and developmental plasticity of their feeding structures[64]. Thus, regulatory mechanisms with a strong stochastic component and environmental influence, might be more frequent regulatory paths in nematodes than previously considered.

## Methods

### Karyotyping

Karyotyping of Prophase I cells (Diakinesis) and gamete cells (sperm or spermatozoa) was performed using live staining of male gonads, as previously described[27]. Briefly, several males were dissected in sperm salt solution (50 mM PIPES, 25 mM KCl, 1 mM $MgSO_4$, 45 mM NaCl, 2 mM $CaCl_2$, pH = 7) with 200 μM Hoechst 33342 on Superfrost Plus glass slides, mounted with coverslips, and observed under a fluorescent microscope (Imager.Z1, Zeiss). The sample numbers are shown in Supplementary Data 1. At least eight individuals were analysed for karyotyping. Because one individual often does not have many countable cells, all observed cells were merged in the statistics (Supplementary Data 1). Chromosome counts can vary among cells. The mode of chromosome numbers in Prophase I cells was determined as haploid chromosome numbers. Gamete chromosome numbers were observed to confirm the haploid numbers and to determine if the sex chromosome system is XO. If less than 60% of sperm cells had the haploid chromosome number, and more than 40% of sperm cells had the haploid chromosome number minus one, the karyotype is categorized as XO system ($n$:$n-1 \leq 60\%$: $> 40\%$). Note that a more relaxed threshold to detect XO systems (e.g. $n$: $n-1 \leq 70\%$: $> 30\%$) led to the same results.

### De novo assembly of *P. fissidentatus*, *P. mayeri* and *P. entomophagus*

For PacBio sequencing, genomic DNA of *P. fissidentatus* (RS5133), *P. mayeri* (RS5460) and *P. entomophagus* (RS144) was extracted from animals using approximately 300 starved plates and the QIAGEN Genomic DNA Maxi kit. Sample preparation for PacBio HiFi sequencing of *P. fissidentatus* was performed using SMRTbell Express Template Prep Kit 2.0, following the manufacturer's instruction (101-853-100 version 1). The sample was sequenced in a half of single molecule real-time sequencing (SMRT) cells of the PacBio Sequel II. Sequencing samples for *P. mayeri* and *P. entomophagus* were prepared and each sequenced in one third of SMRT cell of PacBio Sequel II (HiFi/ CCS mode) by Novogene. PacBio HiFi reads were assembled using the Canu[65] (ver. 2.2) assembler with parameters, genomeSize = 200 m, minReadLength = 10,000, minOverlapLength = 1000 and -pacbio-hifi.

For Hi-C scaffolding, the Hi-C libraries of the three species were prepared from approximately 1000 worms each, using an Arima-HiC kit and a Collibri ES DNA library prep kit, following the manufacturers' instruction. Libraries were sequenced using a MiSeq instrument with the MiSeq reagent kit v3 (101 cycles ×2). For *P. fissidentatus*, *P. mayeri* and *P. entomophagus*, we obtained 5.0 million pairs (0.96Gbp), 7.4 million pairs (1.4Gbp) and 4.9 million pairs (-0.93Gbp) of short reads from Miseq, respectively. Of these, 5.0 million pairs (-99%), 7.3 million pairs (-98%) and 4.7 million pairs (-97%) successfully mapped to each PacBio assembly using BWA-MEM[66] (v0.7.17), respectively. The mapped reads were processed and filtered with the Arima-HiC MappingPipeline (v02) (https://github.com/ArimaGenomics/mapping_pipeline). Finally, we used 2.4 million pairs (433 Mb, -1.7X coverage), 2.5 million pairs (466 Mb, -1.8× coverage) and 3.9 million pairs (721 Mb, -2.5 coverage) of reads for the analysis of *P. fissidentatus*, *P. mayeri* and *P. entomophagus*, respectively. Based on the Hi-C read information, the assembles were further scaffolded using Juicer[67] (ver. 1.6) with default options and the 3D-DNA pipeline[68] (v180114) with option -e. Chromosome-level scaffolds were extracted through manual curation using Juicebox[69] v1.11.08.

After removing scaffolds with BLAST hits of common contaminants from NCBI (ftp://ftp.ncbi.nlm.nih.gov/pub/kitts/contam_in_euks.fa, *e* value < 1e-25), we conducted comparative synteny analysis of those assemblies with the *P. pacificus* genome (El_paco) using nucleotide-based homology detected by LASTZ[70]. We ran LASTZ with the notransition and nogapped options and step = 20 to search for homologous sequences of two genome assemblies. Only homology at unique sites in the new assembly was selected. The largest scaffolds (7 in *P. fissidentatus*; 6 in *P. mayeri* and *P. entomophagus*) were oriented and named based on their correspondence to *P. pacificus* chromosomes, with some exceptions such as in the case of the fused chromosomes. Here, we added the prefixes, 'Pp,', 'Pex,', 'Pf,', 'Pm,' and 'Pen,' to the chromosome names, which stand for *P. pacificus*, *P. exspectatus*, *P. fissidentatus*, *P. mayeri*, respectively.

Evidence-based gene annotation was conducted with the Perl Packages for Customized Annotation Computing annotation pipe line[71] (version 1.0), using previously reported transcriptome data from mixed culture under the laboratory conditions[9].

### Comparative synteny analysis

For comparative synteny analysis, orthologous genes were identified through pairwise reciprocal blastp searches in all protein sequences of the species using NCBI BLAST+[72] (ver. 2.14.0). When a pair of the genes were reciprocally the top hit (i.e. the highest bit score) in the blastp search, genes were considered as orthologues. For *P. mayeri*, *P. entomophagus* and *P. fissidentatus*, the protein sequences of newly annotated genes were used. For the *P. pacificus* and *P. exspectatus*, the latest version of the reference protein sequences were employed

(*P. pacificus*, 'El paco version 3'[71]; *P. exspectatus*, 'exspectatus Yoshida et al.'[27]). Reference protein sequences for the outgroup species were retrieved from WormBase Parasite (WBPS18).

For confirmation of the correspondence to Nigon elements defined in previous studies[28,29], we tested the *P. fissidentatus* chromosomal positions of othogroup genes previously assigned to Nigon elements. First, we downloaded the orthogroup data from the github site (https://github.com/tolkit/otipu_chrom_assem/blob/master/analyses/orthoFinder)[28]. Next, *C. elegans* orthogroup genes were searched in our list of one-to-one ortholog genes between *P. fissidentatus* and *C. elegans*. *P. fissidentatus* chromosomal positions and assigned Nigon elements are compared for the detected orthogroup genes.

To compare the gene density patterns and synteny, sliding window analyses of gene density were performed with non-overlapping 100 kb windows using custom Perl and R scripts. The number of genes per 100 kb window was calculated.

## Coverage analysis

Worm individuals were allowed to crawl on a bacteria-free nematode-growth-medium (NGM) plate for at least 1 h. Subsequently, single worms were lysed in a buffer designed for single-worm lysis (10 mM Tris-HCl, 50 mM KCl, 2.5 mM MgCl$_2$, 0.45% NP-40, 0.45% Tween, 0.2 mg ml$^{-1}$ Proteinase K, pH = 8.3). The library for single-worm whole genome sequencing was prepared as previously described[27]. In brief, we purified the genomic DNA using SpeedBeads magnetic carboxylate modified particles, tagmented it with adaptors using Illumina Tagment DNA Enzyme and Buffer Small Kit, and amplified it through a PCR reaction with the i5 and i7 primers to incorporate barcodes. DNA fragments ranging from 300 to 600 bp were selected using the beads. Genomic DNA, pooled at the same molarity, was then sequenced using Illumina short-read sequencers.

Reference-based coverage analysis was conducted as follows: the sequenced short reads were mapped to the reference sequence using BWA-MEM. The coverage depth of each nucleotide was determined using Samtools[73] (ver. 1.1.0). Sliding window analysis of coverage depth was performed with 100 kb windows without overlap using custom perl and R scripts. The coverage values of a given sample were normalized relative to the autosomal coverage of the same sample and relative to the mean coverage of all female samples. Specifically, the normalized log$_2$ scale coverage, $Y_i$ (y-axis of Figs. 3a–c, 6a) was calculated as:

$$
\begin{aligned}
Y_i = {} & \log_2(\text{average coverage of a 100kb window in the individual } i) \\
& - \log_2(\text{average coverage of autosomes in the individual } i) \\
& - 1/n_f \sum_{j}^{n_f} \log_2(\text{average coverage of a 100kb window of female } j) \\
& + 1/n_f \sum_{j}^{n_f} \log_2(\text{average coverage of autosomes of female } j)
\end{aligned} \quad (1)
$$

where $n_f$ is the sample number of females.

To analyse coverage of each chromosome region in sequence reads from species without a reference sequence, we conducted reference-free coverage analysis. First, we identified conserved 21-mers specific to particular chromosome among *P. pacificus* and *P. fissidentatus*. Then, we counted these 21-mer in the short sequence reads from each individual. The detailed pipeline is as follows: It employed jellyfish[74] ver. 2.3.0 for the following jellyfish commands. Initially, a sequence file for each of seven conserved chromosome elements (i.e. Nigon elements) of *P. pacificus* or *P. fissidentatus* was individually prepared from the genome assembly. *P. pacificus* ChrI was divided at 21.05 Mb into ChrIL and ChrIR, corresponding to NigonE and NigonN, respectively. The other small scaffolds in the assemblies were omitted. Subsequently, 21-mers of each chromosome were counted in the genome assembly of *P. pacificus* and *P. fissidentatus* using jellyfish count with options, -m 21 -s 1 G -C, resulting in jf file, {C}_{S}_all_count.jf, where the parenthesis with C and S represents *P. fissidentatus* chromosome name and species name (i. e. *P. pacificus* or *P. fissidentatus*), respectively. Additionally, 21-mers occurring once in each chromosome were also counted with additional options, -U 1 -L 1, resulting in {C}_{S}_single_copy_count.jf. Conserved 21-mers that are shared between corresponding chromosomes of the two species were selected using jellyfish merge applied to jf file {C}_{S}_single_copy_count.jf with option -U 2 -L 2, resulting in {C}_conserved_single_copy_count.jf. Next, any non-specific 21-mer was removed through a two-step process of jellyfish merge: i) merging the resultant {C}_conserved_single_copy_count.jf and {C}_{S}_all_count.jf of all other chromosomes with option -U 2 -L 2, resulting in {C}_specific_conserved_single_copy_count.jf. ii) merging the {C}_conserved_single_copy_count.jf and {C}_specific_conserved_single_copy_count.jf. with option -U 4 -L 4, resulting in {C}_specific_conserved_single_copy_count_clean.jf. This file was used to calculate the 21-mer coverage in the Illumina reads of whole genome sequencing data of individual samples using the jellyfish count, merge and dump commands. In this analysis, *k*-mers with a coverage greater than 256, likely derived from repeat elements, were removed. The average coverage of each chromosome was normalized against the average coverage of autosomal regions (i.e. genomic region without NigonX). The fold difference in the normalized coverage between randomly selected male and randomly selected female in chromosome *i*, $X_i$ (*x*-axis of Fig. 5a) was calculated as follows:

$$
\begin{aligned}
X_i = {} & \log_2(\text{average coverage of chromosome element } i \text{ in the male}) \\
& - \log_2(\text{average coverage of autosomes in the male}) \\
& - \log_2(\text{average coverage of chromosome element } i \text{ in the female}) \\
& + \log_2(\text{average coverage of autosomes in the female})
\end{aligned}
$$

$$(2)$$

All individuals were used only once to calculate this value.

## Sex ratio test at different temperatures

For sex ratio tests of *P. mayeri* RS5460B, a single P0 hermaphrodite at the J4 juvenile stage was isolated from the laboratory culture. The F1 self-progeny were then raised under normal laboratory conditions (20 °C) until the J4 juvenile stage. Subsequently, three F1 hermaphrodites at the J4 stage were placed on separate plates in an incubator set to the respective temperature. Three test temperatures were simultaneously assessed (15 °C, 20 °C and 25 °C). The F2 self-progeny were raised under the test conditions. Fifteen F2 hermaphrodites at J4 stage were isolated from each of the test conditions. We observed F2 reproduction until animals stopped producing any progeny for two consecutive days. The F3 progeny were cultivated until the adult stage (4–5 days after spawning) to identify their sex (the average number of the F3 progeny: 25 °C, 121.4; 20 °C, 218.1; 15 °C, 215). For the second strain of *P. mayeri* (RSA035) and for *P. entomophagus* RS0144B, the same protocol was applied but only 10 F2 hermaphrodites were tested in each of the conditions (the average number of the F3 progeny of RSA035: 25 °C, 68.4; 20 °C, 160.3; 15 °C, 194.3; the average number of the progeny of RS0144B: 20 °C, 306.9; 15 °C, 328.9). *P. entomophagus* F2 hermaphrodites exhibited limited reproduction (<10 progeny) at 25 °C, and consequently, the 25 °C condition was not used for *P. entomophagus*. Statistic tests were conducted with pairwise likelihood ratio tests of GLMMs, implemented through the glmer function with family = 'binomial' in R. These tests assessed the goodness-of-fit of a test model that included temperature as an explanatory variable, compared to an intercept-only model. The dispersion between different cultural plates was taken into account as a random effect.

## Crossing experiments

As previously described[30], *P. mayeri* RSA035 has one nucleotide change from the strain RS5460 in the rDNA short subunit (SSU) sequence. Thirty J4 virgin juvenile hermaphrodites of RS5460 were isolated and four days later, 10 such adult hermaphrodites (J4+4d), which were depleted of sperm, were crossed with RSA035 males. For each cross, one J4+4d hermaphrodite and one male were placed on a new NGM plate with 50 μl OP50, and parents were transferred to a new plate with 50 μl OP50 every two days until they ceased reproduction. After progeny grew to adulthood, the sex was identified by morphology, and progeny were lysed to analyse the SSU sequence, which was amplified with RH5401 and RH5403 primers (Supplementary Table 2) and Sanger sequenced. Progeny carrying one copy of the RSA035 allele were considered as cross progeny. No progeny indicated developmental impairment.

## Identification of candidate orthologs related to sex determination

To identify candidate orthologs for a sex determination system, orthologous relationships among four *Pristionchus* species, *Parapristionchus givlindavisi*, two *Caenorhabditis* species, *Oscheius tipulae*, and *Strongyloides ratti* were established by Orthofinder[75] v2.5.4. Except for the three species assembled in this study, protein sequences were retrieved from pristionchus.org or WormBase Parasite (WBPS18). The longest isoforms were extracted based on gene IDs using a custom script. Orthofinder was executed with the '-X' option. Based on the Orthofinder outputs, ortholog groups related to sex determination of *C. elegans* were determined.

## CRISPR engineering of *P. mayeri*

The CRISPR guide RNA for *Pmay-tra-1.1* was designed in the conserved zinc finger domain of *Pmay-tra-1.1* (Supplementary Table 2, Fig. 4b). CRISPR engineering was conducted using young adults of *P. mayeri* with a recently improved protocol[36]. We used a co-injection RFP marker with the *P. pacificus eft3* promoter (ChrV: 7,649,953-7,649,357 in *P. pacificus* El_paco genome). This promoter region is 79.8% identical between *P. pacificus* and *P. mayeri*. Mutations were confirmed by sequencing PCR products with primers, Pmay-tra-1.1F and Pmay-tra-1.1R (Supplementary Table 2). Out of 40 injected P0, three lines had marker-positive animals and already had high male ratio in the F1 generation (10–22%). The three mutant lines were established as *tu1863*, *tu1864* and *tu1865*, and had a 2-bp deletion, 5-bp insertion and 5-bp deletion, respectively. The first two lines were used in the following experiments. To test for mating function, mutant males and control wild-type males were crossed with J4+4d adult hermaphrodites, which had depleted their sperms. We tested the presence/absence of F1 progeny (all hermaphrodites) when the hermaphrodites were six and eight days post-reaching the adult stage (J4+6d and J4+8d).

## Reporting summary

Further information on research design is available in the Nature Portfolio Reporting Summary linked to this article.

## Data availability

The data of raw sequence reads and genome assembly generated in this study have been deposited in the NCBI or DDBJ under Project accessions codes, PRJDB16629- PRJDB16631, PRJDB16724-PRJDB16728, and Assembled Data Accessions codes, BTSK01000001-BTSK01000073, BTSX01000001-BTSX01000172, and BTSY01000001-BTSY01000372. The information of source data used in figure generation is provided in Source Data. The other raw data used in this study are available in the figshare database, https://figshare.com/projects/Datasets_and_scripts_for_P_mayeri_sex_determination_project/178452. Source data are provided with this paper.

## Code availability

Scripts are also available in the figshare database, https://figshare.com/projects/Datasets_and_scripts_for_P_mayeri_sex_determination_project/178452.

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

## Acknowledgements

We thank Dr. C. Rödelsperger for helpful comments on the manuscript and support for the bioinformatics. This work was supported by the Max Planck Society and JST CREST grant number JPMJCR18S7 to K.Y.

## Author contributions

K.Y. and R.J.S. designed the study. H.W. performed injections experiments for CRISPR engineering. S.S. and T.K. conducted Hi-C scaffolding. R.H. conducted ortholog analysis for sex determination. W.R. prepared PacBio HiFi sequence library. K.Y. conducted the other experiments and analysis. K.Y. and R.J.S. wrote the manuscript with input from others.

## Funding

## Competing interests

The authors declare no competing interest.
