## [Transparent Peer Review file · Nature Communications]

Rapid chromosome evolution and acquisition of thermosensitive stochastic sex determination in nematode androdioecious hermaphrodites

Corresponding Author: Dr Kohta Yoshida

Version 0:

Reviewer comments:

Reviewer #1

(Remarks to the Author)

Yoshida et al. present a detailed analysis of chromosome evolution and sex determination in the genus *Pristionchus*. The manuscript is the result of an impressive amount of work and reveals surprisingly rapid karyotype evolution in *Pristionchus*, with frequent X-autosome fusion events that have resulted in neo-XX/XY sex chromosome systems in many species. The authors also identify two instances of the evolution of stochastic sex determination. The manuscript is generally well-written and the figures are clear. I commend the authors for making their genome data available via NCBI, which facilitated my review of their manuscript.

However, as I detail below, I do not believe that the major claim of the manuscript - that rapid karyotype evolution is linked to rapid evolution of sex determination mechanisms (and, in turn, androdioecy) - is supported by their data.

Firstly, the authors conflate sex chromosome systems with sex determination mechanisms. In particular, they assume that *Pristionchus* species with the XX/XY sex chromosome systems have sex determination mechanisms that are distinct from their XX/XO ancestors (i.e that the Y chromosome plays a role in sex determination), e.g.:

Lines 42-43 (abstract): "abolished the ancestral XX/XO sex determination system"

Lines 49-50 (abstract): "stochastic sex determination to be derived from XY sex determination"

Lines 50-51 (abstract): "Labile sex determination mechanisms"

Line 125 (introduction): "derived from gonochoristic ancestors with XY sex determination."

Line 127 (introduction): "and a variety of sex determination mechanisms"

Lines 141-142 (results): "highly diverse pattern of chromosome evolution and likely associated sex determination mechanisms"

Lines 156-17 (results): "unexpectedly frequent chromosome evolution and sex chromosome turnover in *Pristionchus* suggesting different sex determination mechanisms"

Line 340 (results): "XY sex determination system"

Lines 369-370 (results): "karyotype evolution in *Pristionchus* is associated with a turnover of primary sex determination mechanisms"

Lines 393-395 (results): "Taken together, our analysis of multiple androdioecious *Pristionchus* species indicates pervasive karyotype and chromosome evolution that is associated with a diversity of sex determination mechanisms."

Lines 439-440 (discussion): "the diversity of sex determination mechanisms in *Pristionchus*"

However, this is highly unlikely to be the case. As is well documented in nematodes and other lineages, XX/XY systems can evolve from XX/XO systems through X-autosome fusion events, where the Y chromosome represents the copy of the former autosome that hasn't fused to the X, as in all XX/XY *Pristionchus* species described here. There is no evidence that the Y chromosomes in these systems play a role in sex determination. A well-known example is the model organism *Drosophila melanogaster*, which has a XX/XY sex chromosome system formed through X-autosome fusion. Despite having an XX/XY system, sex determination in *D. melanogaster* continues to operate via the ancestral X-chromosome counting mechanism and the Y chromosome is preserved only because it contains genes necessary for male fertility (e.g. "The *Drosophila* Y chromosome is not involved in sex determination but contains genes required for male fertility; an XO fly, therefore, is male

but sterile" Gable and Zarkower (2012) *Curr. Biol.*). Therefore, except in cases of SSD, sex determination in XX/XY *Pristionchus* species likely continues to operate via the ancestral X-chromosome counting mechanism described in *C. elegans*. The alternative - that the Y chromosome is somehow involved in sex determination, as in e.g. mammals - would require a highly unlikely scenario where loci on this former autosome repeatedly and independently evolve sex-determining functions. There is therefore no evidence to support the authors' claims that the X-autosome fusion events have "abolished the ancestral XX/XO sex determination system" and contributed to a "variety of sex determination mechanisms" in the genus *Pristionchus*. Note that this issue has been discussed in detail in a recent manuscript on Y chromosome evolution in filarial nematodes, which the authors cite (ref #51, "recent studies suggest that some nematode genera have similar chromosome variation as observed in *Pristionchus*") but otherwise do not discuss further.

Secondly, I do not believe that the authors have established a clear link between the evolution of SSD and karyotype evolution. Although *P. entomophagus* and *P. mayeri* have indeed evolved SSD from an XX/XY ancestor, whether that is coincidental or not is unclear. Given that over half of the *Pristionchus* species studied by the authors are XX/XY (or ancestrally XX/XY), the chance of SSD coincidentally evolving independently in two XX/XY lineages is not particularly low. Indeed, in the only other two nematode species where SSD has been described - *Bursaphalenchus xylophilus* and *Bursaphalenchus okinawaensis* (ref #10 of this manuscript) - there is no link between the evolution of SSD and sex chromosome systems, with SSD evolving within the context of the ancestral XX/XO sex chromosome system. Thus, although it remains possible that sex chromosome evolution predisposes certain species to evolve SSD, the authors do not present convincing evidence that this is the case in *Pristionchus*.

In summary, while the rapid karyotype evolution in *Pristionchus* is interesting, there appears to be very little evidence to suggest that this is related to the evolution of sex determination or androdioecy. The lack of any link is consistent with other statements made in the manuscript, including that "four different karyotypes are found in the androdioecious species" and that the transitions between karyotypes "were not directly associated with reproductive mode".

Minor comments

The authors opt against using Nigon elements to describe karyotypes, despite showing that this elegantly describes chromosome evolution in *Pristionchus*. This means the authors have to rely on pairwise Circos plots to show which chromosomes fused (Figure 2) and on a chromosome naming system that is partly based on *P. pacificus* and partly on *P. fissidentatus*. Both the figures and the text would have been clearer had they used Nigon terminology and showed the distributions of Nigon genes in the various genomes.

Line 170: "150 MB" > "150 Mb"

Line 201: "249Mb" > "249 Mb"

Line 199: "PAC-Bio" > "PacBio"

Line 437: "genome assembly" > "genome assemblies"

Figure 1H: how do the authors think fission has returned the sex chromosome system in *P. americanus* and *P. purgamentorium* back to XX/XO?

Figure 3A-C: the X axis labels are overlapping, a few could be removed to improve readability

Figure 6 legend currently reads "environmental sex determination" (although, based on my concerns above, I do not believe that this figure should remain in the manuscript)

Table 1: "N50" > "Scaffold N50"

Reviewer #2

(Remarks to the Author)

This manuscript shows that there is a higher diversity of sex determination and chromosomal evolution within a nematode clade than previously appreciated. The reported findings change the current view that the XX:XO system with 6 chromosomes is conserved between *C. elegans* and *P. pacificus*, and that the sex chromosome tends to remain unchanged. The authors find variations in sex determination mode (environmental, Y chromosome) and variation in the evolution of the X chromosome within the *Pristionchus* genus. The manuscript is well written and the conclusions are well supported by the experimental data.

I would have just minor suggestions and some additional clarification:

Line 66: add a citation with a more recent review about the topic of transitions between sexual systems in animals:

https://doi.org/10.1007/978-3-319-94139-4_1

Line 141: "androdioecious species are shown in light blue"; however, in Fig. 1H, these species are not labeled in this color. Instead, they are indicated with symbols.

Line 382: it should be Fig. 5C instead of Fig. 5B

Line 385: it should be Fig. 5D instead of Fig. 5CD (or is it meant Fig. 3E, F?)

Figure 1: Using nuclear staining, *P. pacificus* seems to have only ~30% of X-bearing sperm. Is this reflected also in the crosses? Is this lower number of X-bearing sperm a feature of *P. pacificus* of the specific strain tested, or are other strains different? Which strain was used?

Figures 3E and 3F: what is the number of worms scored for calculating the percentages? I could not find the information in the Materials and Methods.

Line 443: *Auanema rhodensis* female/hermaphrodites ratios are not affected by the environment but by the age of the mother (DOI: 10.1038/srep17676). The effect of the environment (social cues) was observed in the species *A. freiburgensis*, affecting the female/hermaphrodite ratios (DOI: 10.1186/s12915-021-01032-1).

Supplementary table: I am assuming that the primers are oriented in the 5'-3' orientation.

Reviewer #3

(Remarks to the Author)

This fascinating paper reveals a host of unexpected evolutionary dynamics in *Pristionchus* nematodes. These include both rampant chromosome fusions and clear examples of non-chromosomal mechanisms of sex determination. The chromosome fusions frequently involve the X, and when they do a stable, unfused neo-X homolog of the X-added region often remains. Such Y chromosomes have been rarely observed in nematodes thus far, adding additional interest to the study. The work is meticulously performed, and the results are rather clear. I can offer a few comments to improve the writing further, but these are relatively minor points, in the order in which they occur in the manuscript (numbers refer to the line of text):

51 The abstract really sells the paper as about androdioecy. However, we actually have a much more general result as noted above, and only two of the seven origins of androdioecy coincide with the evolution of a new karyotype. I therefore object to the phrase "resulted in" here. I suggest this sentence (or something similar) as better capturing the overall results: "Thus, rapid karyotype evolution and labile sex determination mechanisms are a general feature of the genus, and a dynamic background against which androdioecy has evolved recurrently."

72 It's not clear to me what is meant here by "unknown," since we just heard about *C. elegans*, and we already knew about other androdioecious nematodes (some mentioned below). Outside of nematodes, *Eulimnadia* clam shrimp have a longstanding literature. And, as noted below, all are clearly derived from gonochorism. So, "transitions towards androdioecy" are not "largely unknown," but they are indeed relatively few in number. I suggest focusing on this rarity.

111 Be careful here--remember no extant species is "ancestral." Rather, it is an outgroup that likely retains an ancestral character state.

190 A bit more detail in the Methods on how the ortholog assignments were converted into Nigon element diagnoses would be appreciated. How obvious was it?

314 The rapid evolution hypothesis is likely, as it was observed just within *Caenorhabditis* for *xol-1* by Luz et al. (2003) <https://genesdev.cshlp.org/content/17/8/977> and for *fem-3* by Haag et al. (2002) <https://www.sciencedirect.com/science/article/pii/S0960982202013337?via=ihub>

Version 1:

Reviewer comments:

Reviewer #1

(Remarks to the Author)

The authors have extensively reworded their manuscript to clarify the distinction between sex chromosome and sex determination evolution. The additional points in the discussion address my concerns about the link between sex chromosome evolution and androdioecy. The addition of Nigon terminology also helps place their work into the wider context of nematode chromosome evolution. My other minor suggestions have all been addressed.

Reviewer #2

(Remarks to the Author)

The authors have revised the manuscript satisfactorily.

Reviewer #3

(Remarks to the Author)

In this carefully revised MS, the authors have addressed all of this reviewer's concerns, some of which were shared by the other two reviewers.

My only remaining comment relates to line 37 (second line of the Abstract): It's not clear what is meant here by "evolutionary mechanisms toward androdioecy." "Evolutionary mechanisms" could refer to any number of dynamics (e.g. population,

chromosomal, molecular/genic, developmental, etc.) and "toward" is an awkward word choice. So, some kind of rewording is needed. Would the following express the intended point? "...and factors contributing to its evolution remain poorly understood."

RESPONSE TO REVIEWERS' COMMENTS

Please find below our point-by-point response. The comments from reviewers are shown in black. Our responses are shown in green.

Reviewer #1

Yoshida et al. present a detailed analysis of chromosome evolution and sex determination in the genus *Pristionchus*. The manuscript is the result of an impressive amount of work and reveals surprisingly rapid karyotype evolution in *Pristionchus*, with frequent X-autosome fusion events that have resulted in neo-XX/XY sex chromosome systems in many species. The authors also identify two instances of the evolution of stochastic sex determination. The manuscript is generally well-written and the figures are clear. I commend the authors for making their genome data available via NCBI, which facilitated my review of their manuscript.

Response: Thank you for your positive comments. The genome data have been originally uploaded to DDBJ, which automatically synchronizes the data with NCBI, but it might have taken time to synchronize. Now you can see the data in NCBI with the BioProject accessions.

However, as I detail below, I do not believe that the major claim of the manuscript - that rapid karyotype evolution is linked to rapid evolution of sex determination mechanisms (and, in turn, androdioecy) - is supported by their data.

Response: Thank you for this important comment. Our main conclusion is that both karyotype evolution and evolution of (stochastic) sex determination interactively played a role in the evolution of androdioecy as visualized in Figure 6. As the reviewer pointed out, we also think that karyotype evolution resulted in sex chromosome evolution but are not necessarily the driving factor of evolution of “sex determination (mechanism)”. We agree that some representations in our previous manuscript misled this point and could unnecessarily confuse readers. We therefore carefully revised all problematic sentences and clarified the concept adding an explanation in Discussion. (please see details below).

Firstly, the authors conflate sex chromosome systems with sex determination mechanisms. In particular, they assume that *Pristionchus* species with the XX/XY sex chromosome systems have sex determination mechanisms that are distinct from their XX/XO ancestors (i.e that the Y chromosome plays a role in sex determination), e.g.:

Response: We didn't mean to say that the sex determination mechanisms are different between XX/XY and XX/XO systems, but some sentences were falsely represented. Therefore, we revised and changed these statements throughout the manuscript.

Lines 42-43 (abstract): “abolished the ancestral XX/XO sex determination system”

Response: “sex determination system” => “sex chromosome system” (L43)

Lines 49-50 (abstract): “stochastic sex determination to be derived from XY sex determination”

Response: “XY sex determination” => “XY sex chromosome system caused by sex chromosome-autosome fusions” (L49-50)

Lines 50-51 (abstract): “Labile sex determination mechanisms”

Response: We agree with reviewer 1 but also reviewer 3 that these statements were misleading. We have now rephrased the two sentences in the following way: “Thus, rapid karyotype evolution, sex chromosome evolution and evolvable sex determination mechanisms are general features of this genus, and represent a dynamic background against which androdioecy has evolved recurrently.” (L50-53)

Line 125 (introduction): “derived from gonochoristic ancestors with XY sex determination.”

Response: “XY sex determination” => “XY system”. (L127)

Line 127 (introduction): “and a variety of sex determination mechanisms”

Response: “a variety of sex determination mechanisms “ => “the evolution of different sex determination mechanisms” (meaning evolution of stochastic sex determination, L129)

Lines 141-142 (results): “highly diverse pattern of chromosome evolution and likely associated sex determination mechanisms”

Response: “sex determination mechanisms” => “sex chromosome systems” (L144).

Lines 156-17 (results): “unexpectedly frequent chromosome evolution and sex chromosome turnover in *Pristionchus* suggesting different sex determination mechanisms”

Response: We deleted “suggesting different sex determination” and rephrased the sentences as follows: “Together, these findings document an unexpectedly frequent chromosome evolution and sex chromosome turnover in *Pristionchus*. Species without a XO system might have a XY system instead, as *P. expectatus*. However, such a scenario cannot fully explain the karyotype of all species, in particular the four androdioecious species that have lost the XO system. In the following, we use genome sequencing and experimental and reverse genetic approaches to provide first insight into the sex determination mechanisms and the evolution of androdioecy in *Pristionchus*.” (L160-164)

Line 340 (results): “XY sex determination system”

Response: “XY sex determination system” => “XY sex chromosome system”. (L349)

Lines 369-370 (results): “karyotype evolution in *Pristionchus* is associated with a turnover of primary sex determination mechanisms”

Response: We rephrased the sentence as follows: “the sex chromosome evolution predated the evolution of the primary sex determination mechanisms in *P. mayeri*.” (L378-379)

Lines 393-395 (results): “Taken together, our analysis of multiple androdioecious *Pristionchus* species indicates pervasive karyotype and chromosome evolution that is associated with a diversity of sex determination mechanisms.”

Response: We rephrased towards: “Taken together, our analysis of multiple androdioecious *Pristionchus* species indicates pervasive karyotype and sex chromosome evolution, some of which are coupled with the evolution of sex determination mechanisms.” (L403-404)

Lines 439-440 (discussion): “the diversity of sex determination mechanisms in *Pristionchus*”

Response: “the diversity of sex determination mechanisms in *Pristionchus*” => “The parallel evolution of SSD in *Pristionchus*” (L449)

In addition to those points raised by the reviewer, we also reworded as “XX/XO sex determination system” => “XX/XO sex chromosome system” in L93 and “XO sex determination system” => “XO sex chromosome system” in L104.

However, this is highly unlikely to be the case. As is well documented in nematodes and other lineages, XX/XY systems can evolve from XX/XO systems through X-autosome fusion events, where the Y chromosome represents the copy of the former autosome that hasn’t fused to the X, as in all XX/XY *Pristionchus* species described here. There is no evidence that the Y chromosomes in these systems play a role in sex determination. A well-known example is the model organism *Drosophila melanogaster*, which has a XX/XY sex chromosome system formed through X-autosome fusion. Despite having an XX/XY system, sex determination in *D. melanogaster* continues to operate via the ancestral X-chromosome counting mechanism and the Y chromosome is preserved only because it contains genes necessary for male fertility (e.g. “The *Drosophila* Y chromosome is not involved in sex determination but contains genes required for male fertility; an XO fly, therefore, is male but sterile” Gable and Zarkower (2012) *Curr. Biol.*). Therefore, except in cases of SSD, sex determination in XX/XY *Pristionchus* species likely continues to operate via the ancestral X-chromosome counting mechanism described in *C. elegans*. The alternative - that the Y chromosome is somehow involved in sex determination, as in e.g. mammals - would require a highly unlikely scenario where loci on this former autosome repeatedly and independently evolve sex-determining functions. There is therefore no evidence to support the authors’ claims that the X-autosome fusion events have “abolished the ancestral XX/XO sex determination system” and contributed to a “variety of sex determination mechanisms” in the genus *Pristionchus*. Note that this issue has been discussed in detail in a recent manuscript on Y chromosome evolution in filarial nematodes, which the authors cite (ref #51, “recent studies suggest that some nematode genera have similar chromosome variation as observed in *Pristionchus*”) but otherwise do not discuss further.

Response: We fully agree with reviewer and apologize if our previous writing was misleading. As already indicated above, we do not claim that the XX/XY system has different “sex

determination system” as XX/XO system. We think that our rewording throughout the manuscript has now clarified these issues.

Secondly, I do not believe that the authors have established a clear link between the evolution of SSD and karyotype evolution. Although *P. entomophagus* and *P. mayeri* have indeed evolved SSD from an XX/XY ancestor, whether that is coincidental or not is unclear. Given that over half of the *Pristionchus* species studied by the authors are XX/XY (or ancestrally XX/XY), the chance of SSD coincidentally evolving independently in two XX/XY lineages is not particularly low. Indeed, in the only other two nematode species where SSD has been described - *Bursaphelenchus xylophilus* and *Bursaphelenchus okinawaensis* (ref #10 of this manuscript) - there is no link between the evolution of SSD and sex chromosome systems, with SSD evolving within the context of the ancestral XX/XO sex chromosome system. Thus, although it remains possible that sex chromosome evolution predisposes certain species to evolve SSD, the authors do not present convincing evidence that this is the case in *Pristionchus*.

Response: We understand the criticism and concern of this reviewer and have changed our discussion accordingly. We now added the following statement at the end of the discussion: Together, these findings suggest that the evolution of androdioecy in *Pristionchus* nematodes can occur in a context of variations in sex chromosome and sex determination systems. However, it is important to note that the link between the evolution of SSD and karyotype evolution might be coincidental. Although *P. entomophagus* and *P. mayeri* have indeed evolved SSD from an XX/XY ancestor, whether this is causative remains currently unknown. Importantly, in this study, we did not observe any SSD in species having the ancestral 7 chromosomes. In contrast, two independent evolutionary events to SSD were observed after the sex chromosome-autosome fusions. Interestingly, *Bursaphelenchus* species that gained the SSD also experienced sex chromosome-autosome fusions in their ancestor and have six chromosomes (Fig. S2E). However, the total number of examples of SSD in nematodes is too small to draw any final conclusions about necessary predispositions of sex chromosomes. Therefore, the evolutionary relationship between sex chromosome evolution and SSD can only be clarified once more SSD cases are documented. (L503-516)

In summary, while the rapid karyotype evolution in *Pristionchus* is interesting, there appears to be very little evidence to suggest that this is related to the evolution of sex determination or androdioecy. The lack of any link is consistent with other statements made in the manuscript, including that “four different karyotypes are found in the androdioecious species” and that the transitions between karyotypes “were not directly associated with reproductive mode”.

Response: See our comment above. We hope that the revision and new discussion resolve the concern of reviewer 1.

Minor comments

The authors opt against using Nigon elements to describe karyotypes, despite showing that this elegantly describes chromosome evolution in *Pristionchus*. This means the authors have to rely

on pairwise Circos plots to show which chromosomes fused (Figure 2) and on a chromosome naming system that is partly based on *P. pacificus* and partly on *P. fissidentatus*. Both the figures and the text would have been clearer had they used Nigon terminology and showed the distributions of Nigon genes in the various genomes.

Response: We agree that it is helpful to use the Nigon element nomenclature to understand the phenomena in the broader context. Therefore, we first added new quantitative data to compare the *P. fissidentatus* chromosomes and Nigon elements using orthogroup genes previously assigned to Nigon elements. More than 97% of tested genes showed consistency (Table S1). After this clarification and some explanations that were added in the *P. fissidentatus* genome paragraph (L192-200), we now use the Nigon terminology in the rest of the manuscript (e.g. L214-223) and also added information of Nigon elements to Figure 1, 2, 3 & 5.

Line 170: “150 MB” > “150 Mb”

Line 201: “249Mb” > “249 Mb”

Line 199: “PAC-Bio” > “PacBio”

Line 437: “genome assembly” > “genome assemblies”

Response: We corrected all as the reviewer suggested. (L174, 208, 210 446)

Figure 1H: how do the authors think fission has returned the sex chromosome system in *P. americanus* and *P. purgamentorium* back to XX/XO?

Response: This is a very interesting question. The karyotypes of *P. americanus* and *P. purgamentorium* suggest that the fission events happened using the same break point as the fusions and brought them back to XO system. To clarify these events, we think that we need to look at their genomes carefully. Therefore, we did not incorporate any speculation about the fissions into this manuscript.

Figure 3A-C: the X axis labels are overlapping, a few could be removed to improve readability

Response: we removed some labeled numbers.

Figure 6 legend currently reads “environmental sex determination” (although, based on my concerns above, I do not believe that this figure should remain in the manuscript)

Response: We corrected as the reviewer suggested.

Table 1: "N50" > "Scaffold N50"

Response: We corrected as the reviewer suggested.

Reviewer #2

This manuscript shows that there is a higher diversity of sex determination and chromosomal evolution within a nematode clade than previously appreciated. The reported findings change the current view that the XX:XO system with 6 chromosomes is conserved between *C. elegans* and *P. pacificus*, and that the sex chromosome tends to remain unchanged. The authors find variations in sex determination mode (environmental, Y chromosome) and variation in the evolution of the X chromosome within the *Pristionchus* genus. The manuscript is well written and the conclusions are well supported by the experimental data.

I would have just minor suggestions and some additional clarification:

Response: Thank you for the positive comments and productive suggestions below.

Line 66: add a citation with a more recent review about the topic of transitions between sexual systems in animals: https://doi.org/10.1007/978-3-319-94139-4_1

Response: Thank you for alerting us to this review. We have now cited this paper. (L68)

Line 141: "androdioecious species are shown in light blue"; however, in Fig. 1H, these species are not labeled in this color. Instead, they are indicated with symbols.

Response: Yes. This was a mistake. We have now corrected this. (L143)

Line 382: it should be Fig. 5C instead of Fig. 5B

Response: This has been corrected. (L391)

Line 385: it should be Fig. 5D instead of Fig. 5CD (or is it meant Fig. 3E, F?)

Response: Yes. We meant Fig. 5D and corrected accordingly. (L394)

Figure 1: Using nuclear staining, *P. pacificus* seems to have only ~30% of X-bearing sperm. Is this reflected also in the crosses? Is this lower number of X-bearing sperm a feature of *P. pacificus* of the specific strain tested, or are other strains different? Which strain was used?

Response: Actually, yes. We might publish this somewhere in the future. However, this phenomenon is actually strain-specific and so far, we found this trend only in PS312. You can see the extended data fig. 5 of our nature ecology and evolution paper, showing analyses of other strains. (doi.org/10.1038/s41559-022-01980-z)

Figures 3E and 3F: what is the number of worms scored for calculating the percentages? I could not find the information in the Materials and Methods.

Response: We now added the information of the number of progenies used for the test of sex ratio in the materials and method (L675-679).

Line 443: *Auanema rhodensis* female/hermaphrodites ratios are not affected by the environment but by the age of the mother (DOI: 10.1038/srep17676). The effect of the environment (social cues) was observed in the species *A. freiburgensis*, affecting the female/hermaphrodite ratios (DOI: 10.1186/s12915-021-01032-1).

Response: Thank you for this comment. As we understand, the *A. rhodensis* paper also indicated that the environmental stimuli (cholesterol condition and dafachronic acid condition) can change the sex ratio in addition to the maternal age. In the revised manuscript, we wrote: “In *Auanema* species, which exhibit three sexes (i.e. males, females and hermaphrodites), environmental cues and maternal age alter the ratio of females to hermaphrodites in XX animals^{53,54}” and cited both papers (L452-454).

Supplementary table: I am assuming that the primers are oriented in the 5’-3’ orientation.

Response: Exactly. We noted “(5’ to 3’)”.

Reviewer #3:

This fascinating paper reveals a host of unexpected evolutionary dynamics in *Pristionchus* nematodes. These include both rampant chromosome fusions and clear examples of non-chromosomal mechanisms of sex determination. The chromosome fusions frequently involve the X, and when they do a stable, unfused neo-X homolog of the X-added region often remains. Such Y chromosomes have been rarely observed in nematodes thus far, adding additional interest to the study. The work is meticulously performed, and the results are rather clear. I can offer a few comments to improve the writing further, but these are relatively minor points, in the order in which they occur in the manuscript (numbers refer to the line of text):

Response: Thank you for the positive comments and constructive suggestions below.

51 The abstract really sells the paper as about androdioecy. However, we actually have a much more general result as noted above, and only two of the seven origins of androdioecy coincide with the evolution of a new karyotype. I therefore object to the phrase “resulted in” here. I suggest this sentence (or something similar) as better capturing the overall results: “Thus, rapid karyotype evolution and labile sex determination mechanisms are a general feature of the genus, and a dynamic background against which androdioecy has evolved recurrently.”

Response: Thank you for the suggestion. We corrected the sentence in the way that reviewer 3 suggested (L50-53). This also solves a concern of reviewer 1 (see above).

72 It's not clear to me what is meant here by "unknown," since we just heard about *C. elegans*, and we already knew about other androdioecious nematodes (some mentioned below). Outside of nematodes, Eulimnadia clam shrimp have a longstanding literature. And, as noted below, all are clearly derived from gonochorism. So, "transitions towards androdioecy" are not "largely unknown," but they are indeed relatively few in number. I suggest focusing on this rarity.

Response: Thank you for the suggestion. We have now changed the writing towards: "Similarly, empirical studies for evolutionary transitions towards androdioecy are relatively sparse⁵." (L73-74).

111 Be careful here--remember no extant species is "ancestral." Rather, it is an outgroup that likely retains an ancestral character state.

Response: Yes. We corrected as "outgroup" in L113.

190 A bit more detail in the Methods on how the ortholog assignments were converted into Nigon element diagnoses would be appreciated. How obvious was it?

Response: We are happy to follow this advice. In addition to that comment, reviewer 1 suggested to use the Nigon element terminology. Therefore, as indicated above, we newly added the quantitative comparison between *P. fissidentatus* chromosome positions and Nigon elements using orthogroup genes previously assigned to Nigon elements (see a new Table S1 and Materials and Method, L595-601). More than 97% of detected orthogroup genes are located on corresponding *P. fissidentatus* chromosomes.

314 The rapid evolution hypothesis is likely, as it was observed just within *Caenorhabditis* for *xol-1* by Luz et al. (2003) <https://genesdev.cshlp.org/content/17/8/977> and for *fem-3* by Haag et al. (2002) <https://www.sciencedirect.com/science/article/pii/S0960982202013337?via=ihub>

Response: We agree with the reviewer and describe this caveat. Our new wording is: "Additionally, the upstream genes, *xol-1* and *fem-3*, likely experienced rapid protein evolution, which might partially explain why we could not detect some of the genes in species only distantly related to *C. elegans*^{62,63}." in Discussion (L463-465).

Additionally, we found some obvious errors in citations 17 (error in the title) and 47-48 (duplication) and corrected them. Finally, the details of statistic tests are described to complete the check list (L681-685 and Figure 5 legend).

RESPONSE TO REVIEWERS' COMMENTS

There was only one suggestion from the reviewer #3 as,

Reviewer #3 (Remarks to the Author):

In this carefully revised MS, the authors have addressed all of this reviewer's concerns, some of which were shared by the other two reviewers.

My only remaining comment relates to line 37 (second line of the Abstract): It's not clear what is meant here by "evolutionary mechanisms toward androdioecy." "Evolutionary mechanisms" could refer to any number of dynamics (e.g. population, chromosomal, molecular/genic, developmental, etc.) and "toward" is an awkward word choice. So, some kind of rewording is needed. Would the following express the intended point? "...and factors contributing to its evolution remain poorly understood."

We agree the author's suggestion and corrected the first line of the abstract as the reviewer suggested.